# NEDD4 and NEDD4L regulate Wnt signalling and intestinal stem cell priming by degrading LGR5 receptor

Laura Novellasdemunt[1], Anna Kucharska[1], Cara Jamieson[2], Maria Prange-Barczynska[1], Anna Baulies[1], Pedro Antas[1], Jelte van der Vaart[3], Helmuth Gehart[3], Madelon M Maurice[2] & Vivian SW Li[1],*  [ID]

## Abstract

The intestinal stem cell (ISC) marker LGR5 is a receptor for R-spondin (RSPO) that functions to potentiate Wnt signalling in the proliferating crypt. It has been recently shown that Wnt plays a priming role for ISC self-renewal by inducing RSPO receptor LGR5 expression. Despite its pivotal role in homeostasis, regeneration and cancer, little is known about the post-translational regulation of LGR5. Here, we show that the HECT-domain E3 ligases NEDD4 and NEDD4L are expressed in the crypt stem cell regions and regulate ISC priming by degrading LGR receptors. Loss of Nedd4 and Nedd4l enhances ISC proliferation, increases sensitivity to RSPO stimulation and accelerates tumour development in Apc[min] mice with increased numbers of high-grade adenomas. Mechanistically, we find that both NEDD4 and NEDD4L negatively regulate Wnt/β-catenin signalling by targeting LGR5 receptor and DVL2 for proteasomal and lysosomal degradation. Our findings unveil the previously unreported post-translational control of LGR receptors via NEDD4/NEDD4L to regulate ISC priming. Inactivation of NEDD4 and NEDD4L increases Wnt activation and ISC numbers, which subsequently enhances tumour predisposition and progression.

**Keywords** colorectal cancer; intestinal stem cell; Lgr5; NEDD4; Wnt
**Subject Categories** Signal Transduction; Stem Cells & Regenerative Medicine
**The EMBO Journal (2020) 39: e102771**

## Introduction

The intestinal epithelium is constantly self-renewed through a small population of stem cells localised at the bottom of the crypts that continuously regenerate new epithelial cells (Cheng & Leblond, 1974). These ISCs are maintained by a Wnt gradient at the intestinal crypts during homeostasis. Wnt signalling pathway plays central role in multiple cellular processes such as stem cell maintenance and cell fate decision (Clevers & Nusse, 2012). Perturbation of this pathway impairs tissue homeostasis, leading to many diseases including cancer and metabolic disorders (MacDonald *et al*, 2009; Clevers & Nusse, 2012; Novellasdemunt *et al*, 2015). Wnt signalling controls the level of the key effector β-catenin for signal transduction. This is regulated by the cytoplasmic β-catenin destruction complex which consists of the adenomatous polyposis coli (APC), AXIN, glycogen synthase kinase 3 (GSK3) and casein kinase 1 (CK1) (Aberle *et al*, 1997; Kitagawa *et al*, 1999; Liu *et al*, 2002). Wnt ligands initiate the signal transduction through binding to two types of cell-surface receptors: the low-density lipoprotein receptor-related proteins 5 and 6 (LRP5/LRP6) and the Frizzled (FZD) family of serpentine proteins (MacDonald *et al*, 2009; Clevers & Nusse, 2012). Apart from Wnt ligands, the four secreted R-spondin (RSPO) proteins have also been previously reported to enhance canonical Wnt signalling in the presence of Wnt ligands (Kazanskaya *et al*, 2004; Kim *et al*, 2005; Nam *et al*, 2006). Following the discovery of the seven-transmembrane receptor leucine-rich repeat containing G protein-coupled receptor 5 (LGR5) as ISC marker (Barker *et al*, 2007), several studies have further revealed the role of LGR5, and its homologues LGR4 and LGR6, as RSPO receptors for signal enhancement of low-dose Wnt (Carmon *et al*, 2011; Glinka *et al*, 2011; de Lau *et al*, 2011). LGR5 is a Wnt target gene that is enriched in ISCs and colon cancer. It marks adult stem cells in several tissues including the intestinal tract and the hair follicle (Barker *et al*, 2007; Jaks *et al*, 2008), while LGR5[+] cells are also believed to be the cells-of-origin of intestinal cancer (Barker *et al*, 2009). Thus, precise control of LGR5 expression is crucial for stem cell homeostasis and tumour suppression. To date, little is known about the post-translational regulation of LGR5 protein stability.

Several E3 ligases have been previously shown to play important role in ISC maintenance and cancer progression. For instance, the RING finger ubiquitin ligases RNF43 and ZNRF3 have been reported as tumour suppressors and negative regulators of ISC by targeting FZD receptors (Carmon *et al*, 2011; Glinka *et al*, 2011; de Lau *et al*,

1   Stem Cell and Cancer Biology Laboratory, The Francis Crick Institute, London, UK
2   Oncode Institute and Department of Cell Biology, Center for Molecular Medicine, University Medical Center Utrecht, Utrecht, The Netherlands
3   Hubrecht Institute, Royal Netherlands Academy of Arts and Sciences (KNAW) and University Medical Centre (UMC) Utrecht, Utrecht, The Netherlands
    *Corresponding author. Tel: +44 2037 961502; E-mail: vivian.li@crick.ac.uk

2011), while the E3 ligase Mule regulates ISC niche and cancer by targeting EpHB3 and β-catenin (Dominguez-Brauer *et al*, 2016, 2017). On the other hand, the role of Neuronal precursor cell developmentally downregulated protein 4 (NEDD4) and its homologue NEDD4L in cancer is controversial. They belong to the NEDD4 family of HECT-type E3 ubiquitin ligases (Rotin & Kumar, 2009). NEDD4 has been reported as a tumour suppressor (March *et al*, 2011; Zeng *et al*, 2014; Lu *et al*, 2016), while others have suggested that NEDD4 is a proto-oncogene by targeting the tumour suppressor PTEN for degradation (Kim *et al*, 2008; Eide *et al*, 2013). The latter observation was confounded by another study showing that NEDD4 is dispensable for ubiquitination of PTEN in mammalian central nervous system neurons (Hsia *et al*, 2014). Interestingly, NEDD4L has also been shown to inhibit Wnt signalling by targeting Dishevelled (Dvl) for proteasomal degradation (Ding *et al*, 2013).

Here, we investigate the role of NEDD4 and NEDD4L in the context of intestinal homeostasis and tumour development. Through comprehensive analysing of various mouse models, we show that deletion of Nedd4/Nedd4l in the intestinal epithelia increases ISCs and crypt proliferation during homeostasis. Loss of Nedd4/Nedd4l further promotes intestinal tumour progression to high-grade adenomas in Apc$^{min}$ tumour model. We further demonstrate that both NEDD4 and NEDD4L inhibit Wnt signalling by targeting the RSPO receptor LGR5 for lysosomal and proteasomal degradation. Our data unveil the post-translational regulation of LGR5 by the NEDD4 homologues for ISC priming.

# Results

### Loss of Nedd4 and Nedd4l increases ISC numbers and crypt proliferation

We first examined the expression pattern of *Nedd4* and *Nedd4l* in murine intestine. RNAScope *in situ* hybridisation (ISH) showed that *Nedd4* was expressed predominantly at the crypt stem cell zone, while *Nedd4l* was homogenously distributed throughout the crypt–villus axis (Fig 1A). The result was confirmed by quantitative reverse transcription polymerase chain reaction (qRT–PCR) of crypt and villus fractions of intestinal crypts (Fig EV1A). Interestingly, expression of *Nedd4* was significantly upregulated in Apc$^{min}$ adenoma, while *Nedd4l* expression was unchanged (Fig 1B). This is consistent with the previous observation in human colorectal cancer tissues (Tanksley *et al*, 2013). The results suggest that Nedd4 expression could be regulated by Wnt signalling.

To investigate the functional role of Nedd4 and Nedd4l in intestinal homeostasis, we generated Villin$^{CreERT2}$;Nedd4$^{fl/fl}$ (Nedd4 cKO) and Villin$^{CreERT2}$;Nedd4l$^{fl/fl}$ (Nedd4l cKO) mice to induce gene deletion in intestinal epithelia upon tamoxifen administration. We further generated double-mutant Villin$^{creERT2}$;Nedd4$^{fl/fl}$;Nedd4l$^{fl/fl}$ (DKO) mice to delete *Nedd4* and *Nedd4l* simultaneously. Single- or double-mutant intestines were examined 50 days post-induction (dpi), which showed no significant changes in gross intestinal morphology or crypt proliferation (Fig EV1B). Interestingly, when we let the animals age for 1 year, significant increase in crypt proliferation was observed in the DKO intestine (Fig 1C, D and O) with elongated crypts (Fig 1E, F and P), indicating expansion of proliferative crypt compartment. RNAScope ISH analysis of the DKO

intestine further showed increased expression of the ISC marker *Olfm4* (Figs 1G, H and Q). Consistently, the number of Cyclin d1- and Sox9-positive cells was also increased in the DKO intestine (Fig 1I, L, R and S), suggesting that Wnt signalling is upregulated. qRT–PCR analysis further confirmed a significant increase in Wnt target genes and stem cell markers expression in the DKO intestine (Fig EV1C). We did not observe significant changes in Paneth cell numbers (Fig 1M, N and T). Our data indicate that prolonged deletion of *Nedd4* and *Nedd4l* leads to increased crypt proliferation and ISC numbers.

### Nedd4/Nedd4l deficiency activates Wnt signalling and promotes growth advantage in intestinal organoids

To validate the stem cell expansion phenotype, we further examined the intestinal organoids derived from wild-type (WT), Nedd4 cKO, Nedd4l cKO and DKO animals at 7dpi. Loss of Nedd4 or Nedd4l was confirmed by qRT–PCR (Fig EV2A). Surprisingly, significant upregulation of Wnt target genes (Fig 2A) and stem cell markers (Fig 2B) was observed in all mutant organoids at short-term gene deletion (7dpi) as compared to 1 year *in vivo*. There was also a remarkable increase in proliferation (Fig 2C) and organoid formation efficiency in the mutant organoids (Figs 2D and EV2B), supporting the notion that Nedd4 or Nedd4l deletion increases numbers of ISCs. We speculate that the accelerated phenotypes in organoids may be attributed to the growth factor-enriched culture condition *in vitro*. To test whether the growth advantage of the mutant organoids is dependent on exogenous Wnt signal, we further challenged the organoids by reducing the amount of the Wnt agonist RSPO from the culture media, which is an essential component of the organoid media (Sato *et al*, 2009). Neither WT nor mutant organoids survived in the absence of RSPO. On the other hand, in the RSPO-low (1%) condition, most WT organoids died at day 3 while the mutant organoids were able to survive longer (~day 7–8) (Fig 2E and F). The results indicate that loss of Nedd4/Nedd4l results in organoid growth advantage upon RSPO depletion, while the survival of mutant organoids is still dependent on exogenous RSPO.

### Loss of Nedd4 and Nedd4l exacerbates Apc$^{min}$ intestinal tumour phenotype with increased tumour grade

Nedd4 has been previously reported to enhance growth of Apc$^{min}$ intestinal tumours, which required deletion of *Nedd4* in both intestinal epithelium and the surrounding tissues (Lu *et al*, 2016). The role of Nedd4l in colorectal cancer (CRC) remains uncharacterised. To clarify the role of Nedd4 and Nedd4l in intestinal tumorigenesis, we further crossed the mutant mice to Apc$^{min}$ animals, a mouse model of colon cancer (Su *et al*, 1992), to obtain Apc$^{min}$ Nedd4 cKO, Apc$^{min}$ Nedd4l cKO, and the compound Apc$^{min}$ DKO mice. While Villin$^{CreERT2;\ Apcmin}$ (designated Apc$^{min}$ control) mice lived for approximately 5 months (151 days on average), Apc$^{min}$ Nedd4l cKO and Apc$^{min}$ DKO showed signs of sickness significantly earlier at around 4 months (125 and 126 days on average, respectively) (Figs 3A and EV3A). The survival rate of Apc$^{min}$ Nedd4 cKO animals was also decreased but not significant (140 days on average) (Fig EV3A). Apc$^{min}$ mice with single or double deletion of *Nedd4* and *Nedd4l* exhibited an increase in tumour numbers in small intestine, while Apc$^{min}$ DKO animals further displayed increased numbers of colonic

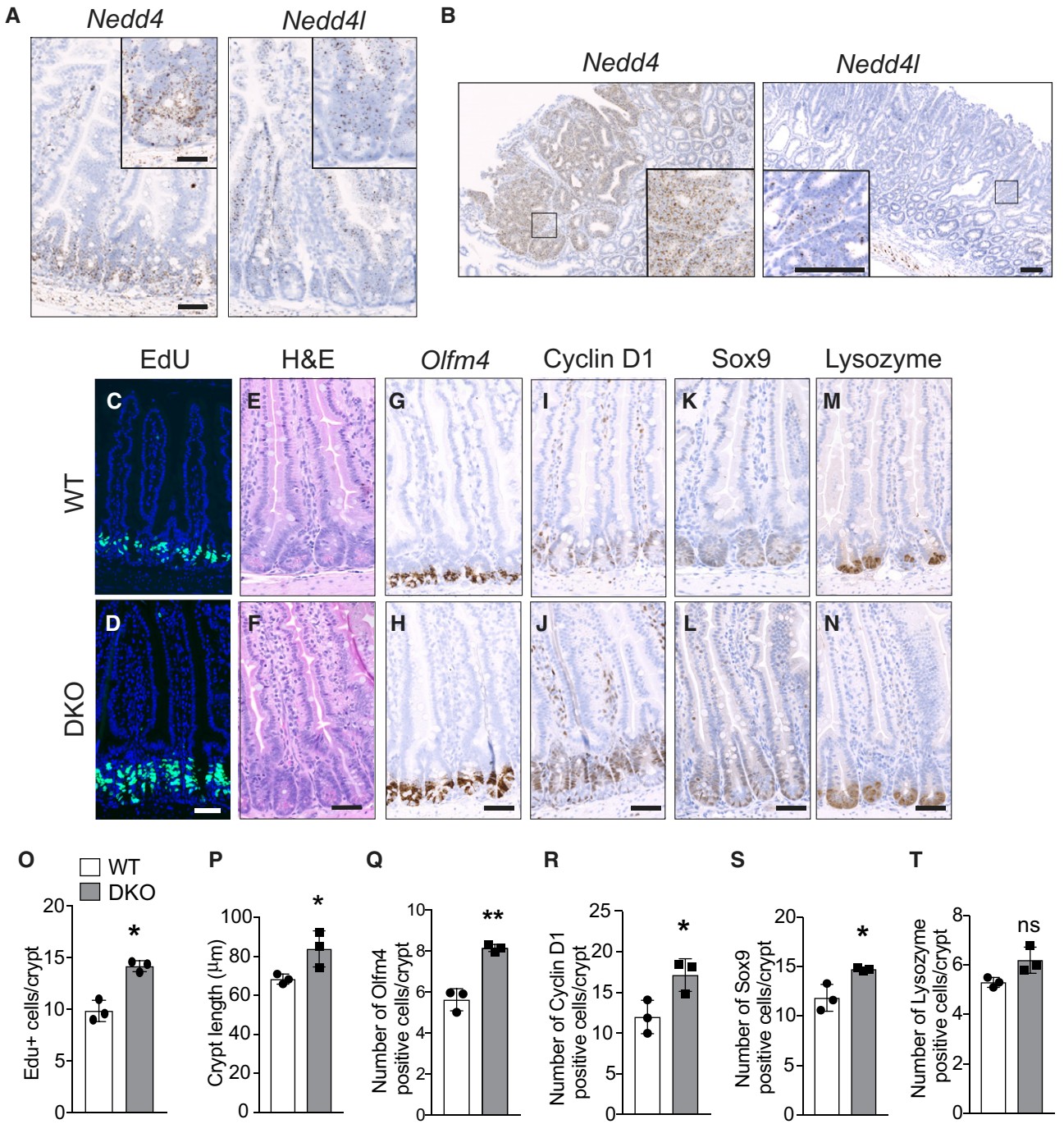

**Figure 1.  Loss of Nedd4 and Nedd4l increases intestinal crypt proliferation.**

A, B　Representative images of RNAScope ISH showing Nedd4 and Nedd4l expression in mouse small intestinal tissues derived from normal animal (A) and Apc[min] tumours (B). High-magnification images are shown in the box. Scale bars, 50 μm (A), 100 μm (B).

C–N　Histology and immunostaining of Villin[creERT2] wild-type (WT) (C, E, G, I, K, M) and Villin[creERT2];Nedd4[fl/fl];Nedd4l[fl/fl] DKO (D, F, H, J, L, N) proximal intestine 1-year post-tamoxifen induction using the indicated markers (*n* = 3 per group). Scale bars, 50 μm.

O　　Quantitation of Edu[+] proliferating cells per crypt from (C, D). Each dot represents the average of at least 10 crypts per animal. Data are mean ± standard error. *n* = 3 per group.

P　　Quantitation of crypt length (μm) from (E, F). Each dot represents the average of at least 20 crypts per animal. Data are mean ± standard error. *n* = 3 per group.

Q–T　Quantitation of Olfm4 (Q), Cyclin D1 (R), Sox9 (S) and lysozyme (T)-positive cells per crypt. Each dot represents the average of at least 20 crypts per animal. Data are mean ± standard error. *n* = 3 per group. Error bars represent ± SEM.

Data information: *P* values were determined using the unpaired two-sided *t*-test (**P* < 0.05; ***P* < 0.01; ns, not significant).

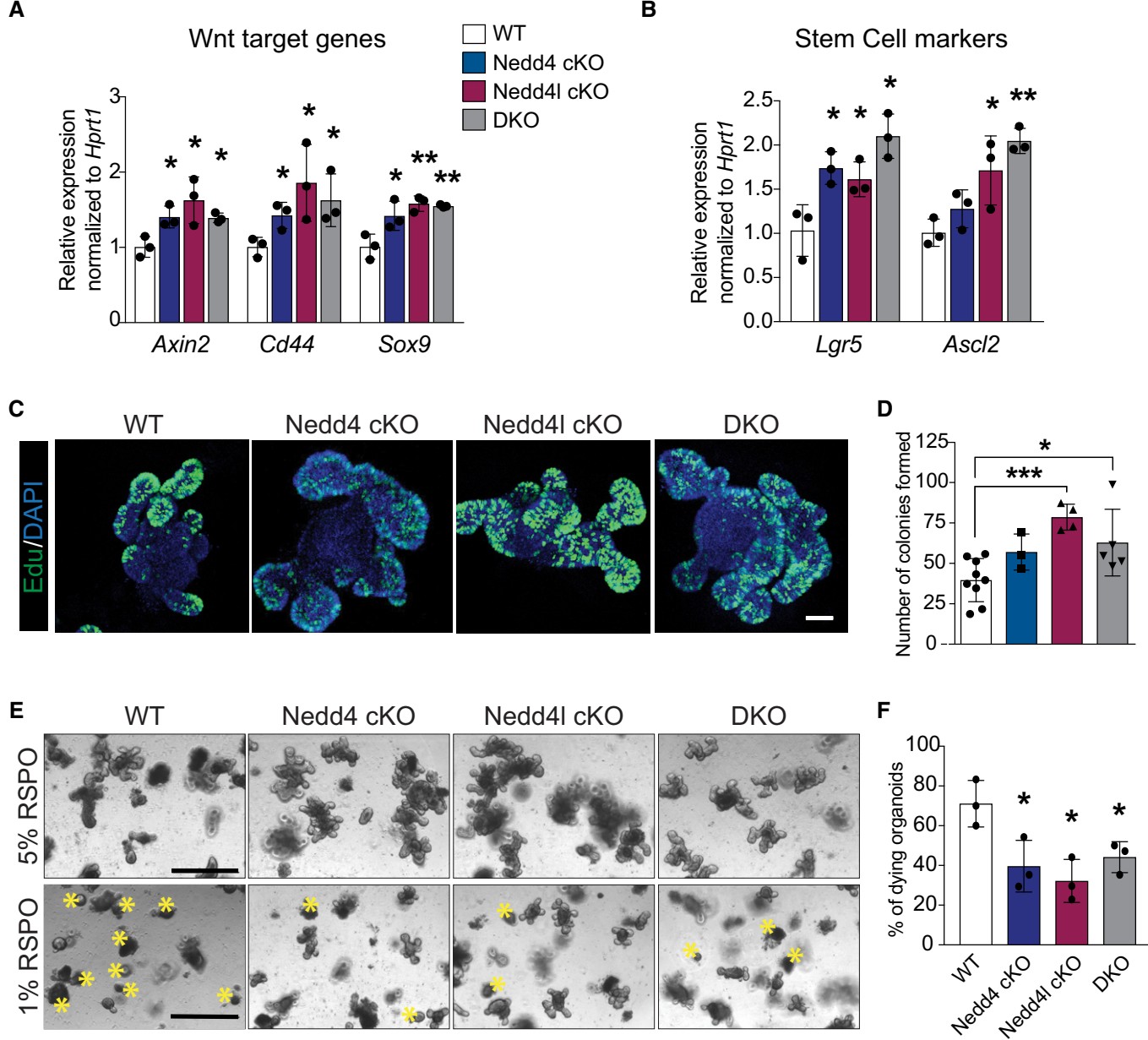

**Figure 2. Nedd4/Nedd4l deficiency activates Wnt signalling and promotes growth advantage in intestinal organoids.**

A, B   mRNA expression of the indicated genes was analysed by qRT–PCR in small intestinal organoids isolated from the correspondent WT, Nedd4 cKO, Nedd4l cKO and DKO mice. Data are presented as fold change normalised to Hprt1 control in triplicate ($n$ = 3 per condition). Error bars represent ± standard error.

C   Immunostaining for proliferating cells after 2-h EdU incorporation. Scale bars, 100 μm.

D   Quantitation of the organoid formation assay in (Fig EV2B). WT = 9, Nedd4 cKO = 3, Nedd4l cKO = 4 and DKO = 5. Data are mean ± standard error.

E   Representative images of organoids of the indicated genotypes cultured under 5% RSPO (top row) or 1% RSPO (bottom row) conditions at day 3. Asterisks indicate dying organoids. Scale bar, 1,000 μm.

F   Quantitation of dying organoids in (E). Each dot represents the average of dying organoids from three different mice per genotype in the indicated conditions. Data are mean ± standard error.

Data information: $P$-values were determined using the unpaired two-sided $t$-test (*$P$ < 0.05; **$P$ < 0.01; ***$P$ < 0.001).

tumours (Fig 3B andD). Histology analysis revealed that most adenomas were low-grade dysplasia, whereas Apc[min] Nedd4l cKO and Apc[min] DKO further promoted high-grade dysplasia (Fig 4A–C). In contrast to the previous study (Lu *et al*, 2016), our data show that deletion of Nedd4 and/or Nedd4l in the intestinal epithelium alone

is sufficient to accelerate adenoma development in Apc[min] animals. Of note, loss of Nedd4l appears to be more effective in tumour progression than Nedd4 ablation.

Since deletion of Nedd4 and Nedd4l increases ISC numbers under homeostasis, we further examined the ISC marker *Lgr5*

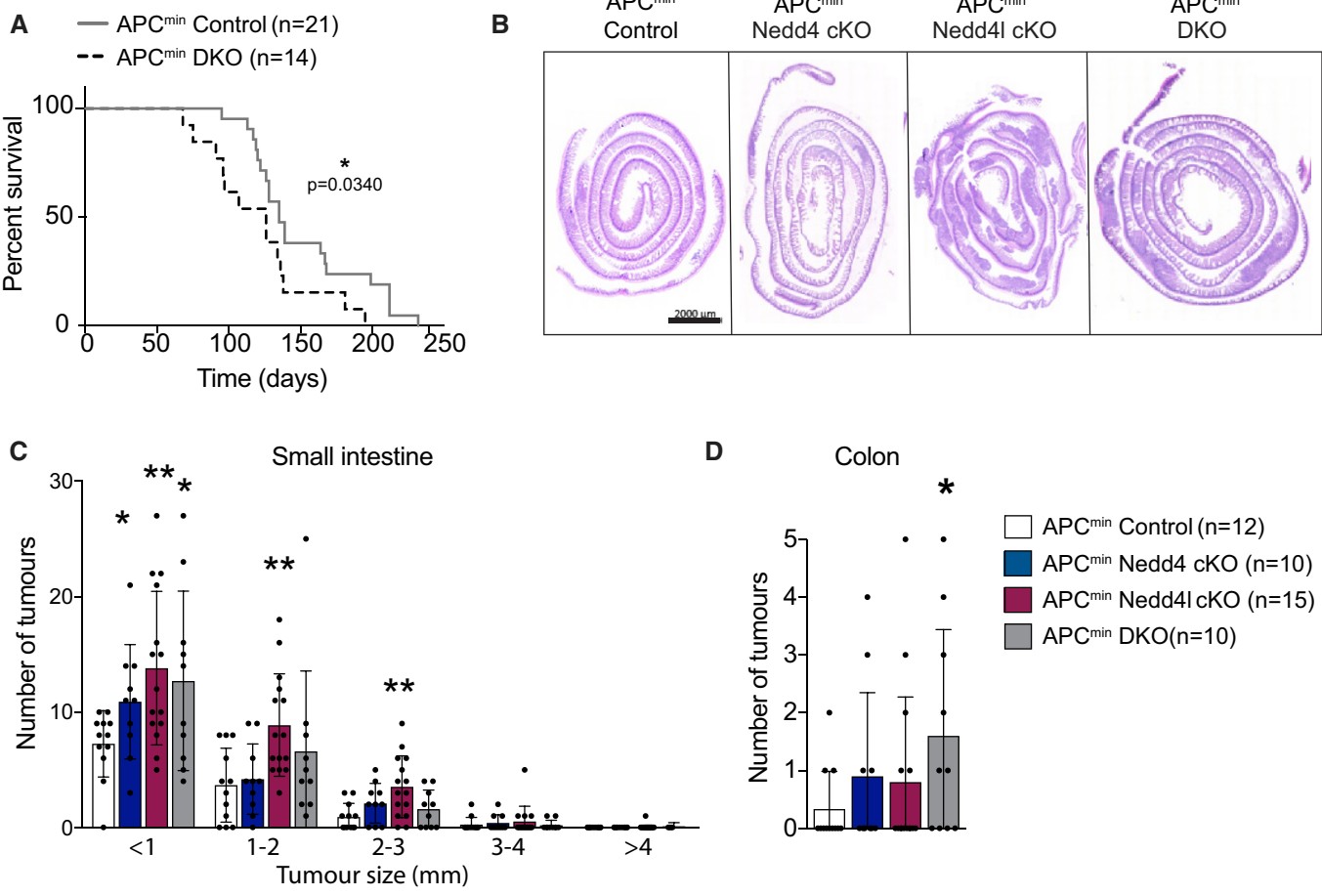

**Figure 3. Loss of Nedd4 and Nedd4l exacerbates Apc^min intestinal tumour phenotype.**

A   Kaplan–Meier survival analysis of Apc^min control and Apc^min DKO mice. *P* values were determined using the Mantel–Cox test.
B   Representative H&E staining of the small intestines of the indicated genotypes. Scale bar 2,000 μm.
C, D   Total number of adenomas in the small intestine (C) and colon (D) 3 months after induced Nedd4 and/or Nedd4l loss. Data are mean ± standard error. *P*-values were determined using the unpaired two-sided *t*-test (*P < 0.05; **P < 0.01).

expression in control and mutant adenomas. Consistently, loss of Nedd4 and/or Nedd4l resulted in an increase in *Lgr5*-expressing stem cells in the adenomas (Fig 4D–G). In addition, the number of lysozyme+ Paneth cells was also significantly increased in mutant adenomas as compared to the Apc^min control (Figs 4H–K and EV3B). Next, we asked if the tumour-promoting role of Nedd4 and Nedd4l was associated with Wnt signalling. Immunohistochemistry analysis showed strong nuclea β-catenin staining in all control and mutant adenomas (Fig EV3C–F). On the other hand, the Wnt target gene Sox9 expression was significantly upregulated in all mutant adenomas when compared to control (Fig EV3G–J and O), suggesting that Wnt signalling is upregulated upon Nedd4 and Nedd4l loss. In addition, adenomas with single or double deletion of *Nedd4* and *Nedd4l* further displayed increased proliferation as indicated by Edu⁺ cells (Figs 4L–O and EV3P), while apoptosis was not affected (Fig EV3K–N). Together, we conclude that loss of Nedd4 and Nedd4l in Apc^min animals promotes intestinal tumour progression by enhancing Wnt activation with increased numbers of ISCs and Paneth cells.

## The E3 ligases NEDD4 and NEDD4L negatively regulate Wnt signalling upstream of the β-catenin destruction complex

To study the Wnt regulatory role, we first deleted NEDD4 or NEDD4L via CRISPR targeting in HEK293T cells. Loss of NEDD4 or NEDD4L protein was confirmed by Western blot analysis (Fig EV4A). This resulted in significant increase in TCF-transcriptional (TOPFlash) activity in all the mutant cells in both Wnt3a and Wnt3a plus RSPO conditions (Figs 5A and EV4B). NEDD4L has been previously reported as Wnt negative regulator by targeting DVL for degradation (Ding *et al*, 2013). We asked if NEDD4 also contributed to DVL degradation. Interestingly, our data showed that overexpression of either NEDD4 or NEDD4L induced the degradation of DVL2 but not DVL1/3 (Figs 5B and EV4C). Immunoprecipitation (IP) analysis further showed that WT NEDD4, but not the catalytic inactive mutant C854S (NEDD4-CS), promoted DVL2 ubiquitination (Fig EV4D). These results indicate that both NEDD4 and NEDD4L play redundant roles in targeting DVL2 for proteasomal degradation.

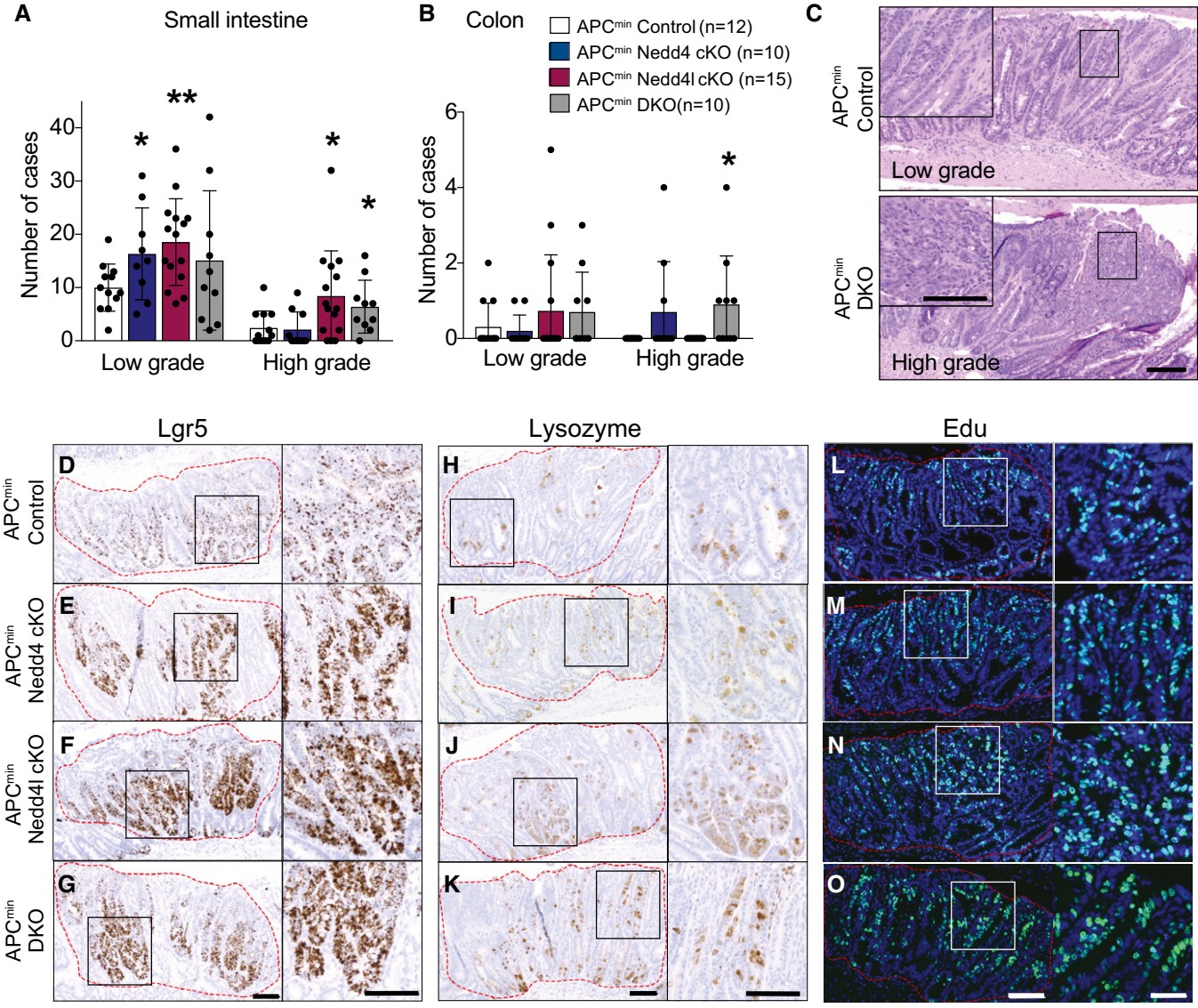

**Figure 4. Inactivation of NEDD4 and NEDD4L increases proliferation and tumour grade in Apc^min mice.**

A, B Quantitation of the grades of the adenomas that developed in the small intestine (A) and colon (B) of the mice in (C, D). Data are mean ± standard error. *P*-values were determined using the unpaired two-sided *t*-test (**P* < 0.05; ***P* < 0.01).

C Representative H&E staining from a low-grade versus a high-grade tumour in the indicated genotypes. Scale bar, 100 μm.

D–O Representative images of Lgr5 RNAScope (D–G), lysozyme (H–K) and Edu (L–O) staining in adenoma tissues obtained from the indicated genotypes. High-magnification images of the boxed areas within the adenomas are shown on the right. Scale bar, 100 μm. Data are representative of at least three mice per group. Adenomas are circled in red.

Since our organoid data showed that Nedd4 and Nedd4l mutants exhibited growth advantage upon RSPO depletion, we asked if Nedd4/Nedd4l also play a role in the RSPO axis of the Wnt pathway. Ectopic expression of either NEDD4 or NEDD4L significantly suppressed the TOPFlash activity induced by Wnt3a and RSPO treatment (Fig 5C). To test whether this Wnt inhibitory role is DVL2-dependent, we further deleted DVL2 in HEK293T cells (Fig EV4E). Surprisingly, NEDD4 and NEDD4L were still able to suppress Wnt activity in the absence of DVL2 (Fig 5D), suggesting that there might be additional player for NEDD4/4L-mediated Wnt inhibition. To further dissect how NEDD4 and NEDD4L regulate

Wnt signalling, we assessed their inhibitory effect using different cell lines carrying different mutations in the Wnt pathway. Similar to HEK293T cells, expression of NEDD4 and NEDD4L in the CRC cell line HCT116 strongly suppressed Wnt pathway activation (Fig 5E). On the other hand, ectopic expression of NEDD4 and NEDD4L in APC4 (HEK293T cells with APC truncation at 1,225a.a.) (Novellasdemunt *et al*, 2017) and DLD1 (APC-mutated CRC cells at 1,417a.a.) cells failed to inhibit Wnt signalling (Fig 5F and G). HCT116 cells have WT APC but heterozygous β-catenin mutation. Importantly, HCT116 cells have been reported to express Wnt ligands for autocrine Wnt signalling (Bafico *et al*, 2004).

These data suggest that NEDD4 and NEDD4L inhibit Wnt signalling upstream of the APC/β-catenin destruction complex, likely at the surface receptor level.

## NEDD4 and NEDD4L target LGR5 receptor for lysosomal and proteasomal degradation

We speculate that NEDD4 and NEDD4L may inhibit Wnt signalling by targeting Wnt receptors for degradation. To test our hypothesis, we examined the effect of NEDD4/NEDD4L expression on the common canonical Wnt receptors FZD4 and FZD5, and the two RSPO receptors LGR4 and LGR5. Our data showed that over-expression of NEDD4 and NEDD4L WT constructs, but not their catalytic inactive mutants (NEDD4-CS and NEDD4L-CA), resulted in a marked reduction of LGR4 and LGR5 protein levels (Figs EV5A and 6A). In contrast, expression of NEDD4 and NEDD4L did not alter the protein levels of FZD4 or FZD5 Wnt receptors (Figs EV5B and 5C). Interestingly, two bands of LGR5-Flag proteins were noted in the immunoblot: the lower band for the immature unprocessed ER form and the upper band for the mature, glycosylated post-Golgi form (Fig 6A). The NEDD4/NEDD4L-mediated degradation appeared to happen predominantly in the upper mature form. To confirm these findings, we co-expressed HEK293T cells with NEDD4/NEDD4L and LGR5 that carried a SNAP-tag (SNAP-LGR5) in its extracellular domain. SNAP-LGR5 labelling with membrane-impermeable SNAP-Alexa488 showed its localisation at the cell surface (Fig 6B). Notably, expression of WT NEDD4 or NEDD4L strongly reduced this cell-surface pool of LGR5 signal, while the signal was largely unaffected in NEDD4-CS and NEDD4L-CA mutant-expressing cells (Figs 6B and EV5E). This is consistent with the immunoblot data that only mature form of LGR5 at the membrane level was being degraded by NEDD4 and NEDD4L, while the total LGR5 level (mostly cytoplasmic as immature form in the biosynthetic pathway such as ER) was unaffected. On the other hand, expression of NEDD4 and NEDD4L did not alter the surface levels of SNAP-FZD5 (Figs EV5D and 5E), which is in concordance with our earlier observation (Fig EV5C). Thus, our data support the notion that NEDD4 and NEDD4L selectively target LGR but not FZD receptors for degradation.

Next, we examined the effect of NEDD4 and NEDD4L on LGR5 protein turnover. Interestingly, LGR5 protein was highly unstable. Significant degradation of total LGR5 protein was detected as soon as 1 h after cycloheximide (Chx) treatment in control cells, which was stabilised in NEDD4 and NEDD4L CRISPR KO cells (Fig EV5F). Collectively, the data suggest that NEDD4/NEDD4L promote LGR5 degradation.

To investigate whether the E3 ligases NEDD4 and NEDD4L regulate LGR5 turnover through ubiquitination-mediated degradation, HEK293T cells were overexpressed with MYC-NEDD4/MYC-NEDD4L, LGR5-Flag and HA-Ubiquitin followed by IP analysis. In the presence of lysosomal inhibitor Bafilomycin A1, expression of NEDD4 or NEDD4L strongly induced ubiquitination of LGR5 (Figs EV5G and 5H). On the other hand, LGR5 ubiquitination efficiency was reduced when co-expressed with the NEDD4-CS or NEDD4L-CA mutants, indicating that NEDD4 and NEDD4L target LGR5 for lysosomal degradation via their E3 ligase activities (Figs EV5G and 5H). To determine whether proteasomal

degradation is also involved, we treated the cells with proteasomal inhibitor MG132 and repeated the IP experiment. Strikingly, expression of WT NEDD4, but not the mutant, resulted in robust accumulation of ubiquitinated form of LGR5 (Fig 6C). Similar results were observed using NEDD4L overexpression (Fig 6D), indicating that both NEDD4 and NEDD4L mediate proteasomal degradation of LGR5. In a reverse experiment, we examined LGR5 ubiquitination in the CRISPR-mediated NEDD4 and NEDD4L KO cells. Loss of NEDD4 or NEDD4L displayed a profound reduction in LGR5 ubiquitination, supporting the notion that LGR5 ubiquitination requires NEDD4/NEDD4L (Fig 6E). Together, our results indicate that NEDD4 and NEDD4L target LGR5 for both lysosomal and proteasomal degradation.

Next, we examined if NEDD4 and NEDD4L interact with their substrates. Co-IP analysis showed that NEDD4 and NEDD4L indeed bound to LGR4 and DVL2 at the endogenous level (Appendix Fig S1A). Consistently, endogenous LGR4 and DVL2 protein levels were stabilised in the NEDD4 or NEDD4L CRISPR KO cells (Appendix Fig S1B), indicating that they are the substrates for the NEDD4 E3 ligases. Unfortunately, we were not able to demonstrate LGR5 stabilisation at the endogenous level due to the lack of reliable antibodies. To validate that the NEDD4/NEDD4L-mediated Wnt regulation is dependent on LGR4/5, we further performed siRNA knockdown (KD) assays in the NEDD4 and NEDD4L CRISPR KO cells. Loss of LGR4/LGR5 abrogated the Wnt activation in the NEDD4/NEDD4L KO cells, indicating that the NEDD4/NEDD4L-mediated Wnt regulation is indeed dependent on LGR4/5 (Fig 6F and Appendix Fig S1C). To further study the endogenous Lgr5 protein level in the absence of reliable antibodies, we introduced HA tag to the endogenous Lgr5 locus (Lgr5-HA tag) in WT and DKO mouse organoids (Appendix Fig S1D). Western blot analysis confirmed the increase in the endogenous Lgr5-HA protein level in the DKO organoids when compared to WT (Appendix Fig S1E). Taken together, these results support the notion that the E3 ubiquitin ligases NEDD4 and NEDD4L target LGR4, LGR5 and DVL2 for degradation.

## Loss of Nedd4 and Nedd4l increases sensitivity to RSPO stimulation

Given our observation that Nedd4 targets the RSPO receptor Lgr5 for degradation, we asked if deletion of Nedd4 and/or Nedd4l *in vivo* will synergise RSPO signalling. Tamoxifen was injected into WT and mutant animals 7 days prior RSPO3 injections for 3 days (Fig 7A). Injection of RSPO3 alone in WT animals resulted in increased crypt proliferation, while deletion of Nedd4 and/or Nedd4l further synergised the RSPO-induced hyperproliferation (Fig 7B and Appendix Fig S2A). Consistently, the population of Oflm4[+] ISCs was further expanded in Nedd4l cKO and DKO intestine when compared to WT upon RSPO3 administration (Fig 7C and Appendix Fig S2B). The results indicate that deletion of Nedd4 or Nedd4l *in vivo* leads to hypersensitivity to RSPO stimulation. It is interesting to note that loss of Nedd4 and Nedd4l alone did not show ISC expansion phenotype until 1 year after deletion *in vivo* (Figs 1 and EV1). In contrast, hyperproliferation of intestinal crypts and ISC expansion were observed as early as 10 dpi under RSPO stimulation, supporting the notion that Nedd4 and Nedd4l regulate RSPO-LGR axis of the Wnt pathway.

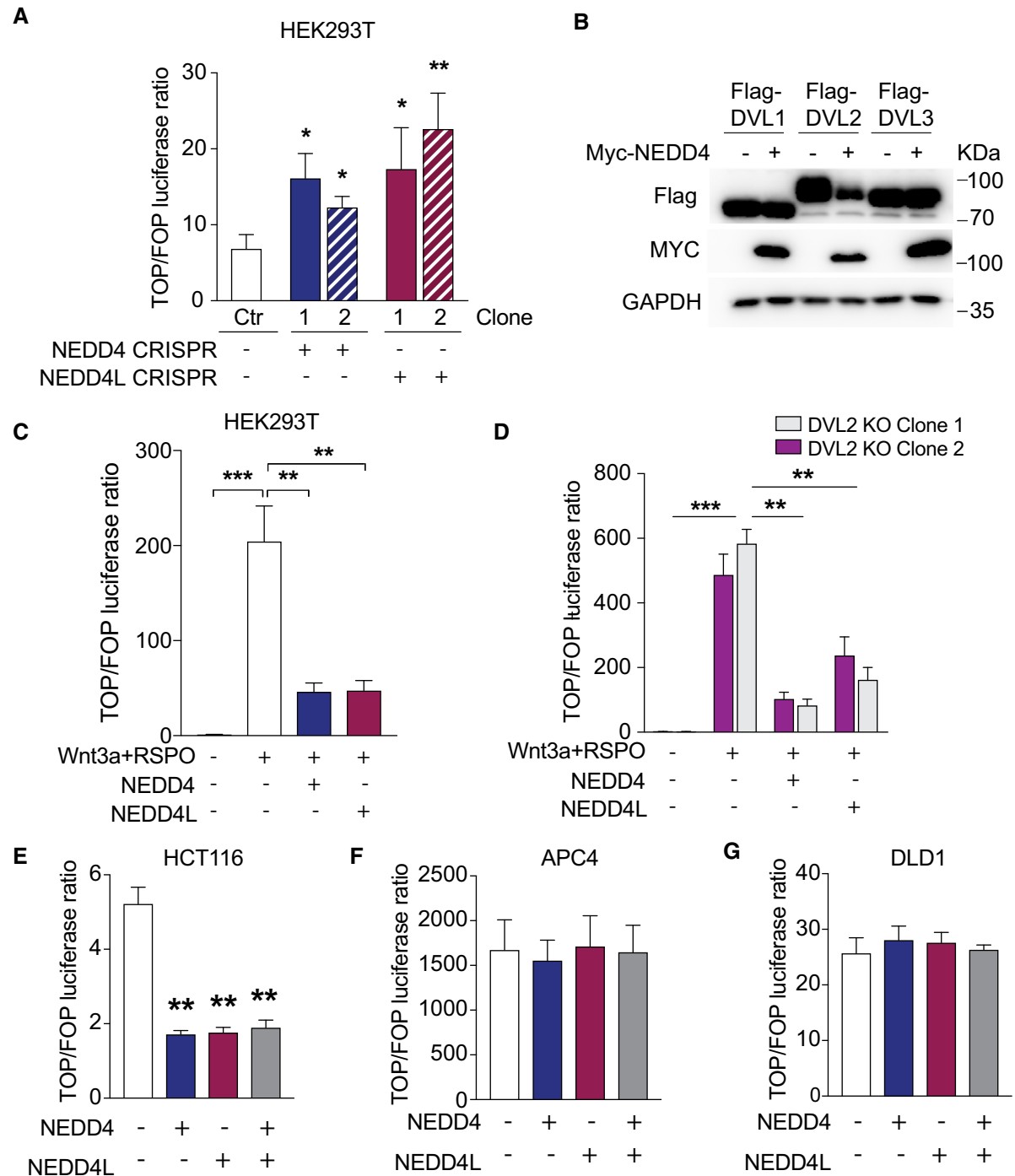

**Figure 5. NEDD4 and NEDD4L negatively regulate Wnt signalling pathway.**

A Relative TOPFlash reporter activities of HEK293T cells with the indicated CRISPR targeting. Cells were treated with Wnt3a-conditioned media.

B Western blot analysis of HEK293T cells transfected with Flag-DVL1, Flag-DVL2 or Flag-DVL3 with or without Myc-NEDD4 using the indicated antibodies.

C Relative TOPFlash reporter activity upon ectopic expression of the indicated plasmids in HEK293T cells treated with Wnt3a and RSPO.

D Relative TOPFlash reporter activities of HEK293T cells with CRISPR deletion of DVL2 upon ectopic expression of the indicated plasmids. Cells were treated with Wnt3a and RSPO.

E–G Relative TOPFlash reporter activities of HCT116 (E), APC4 (Novellasdemunt *et al*, 2017) (F) and DLD1 (G) cells transfected with the indicated plasmids.

Data information: Data represent mean ± standard error of at least three independent experiments. *P*-values were determined using the unpaired two-sided *t*-test (*$P < 0.05$; **$P < 0.01$; ***$P < 0.001$).

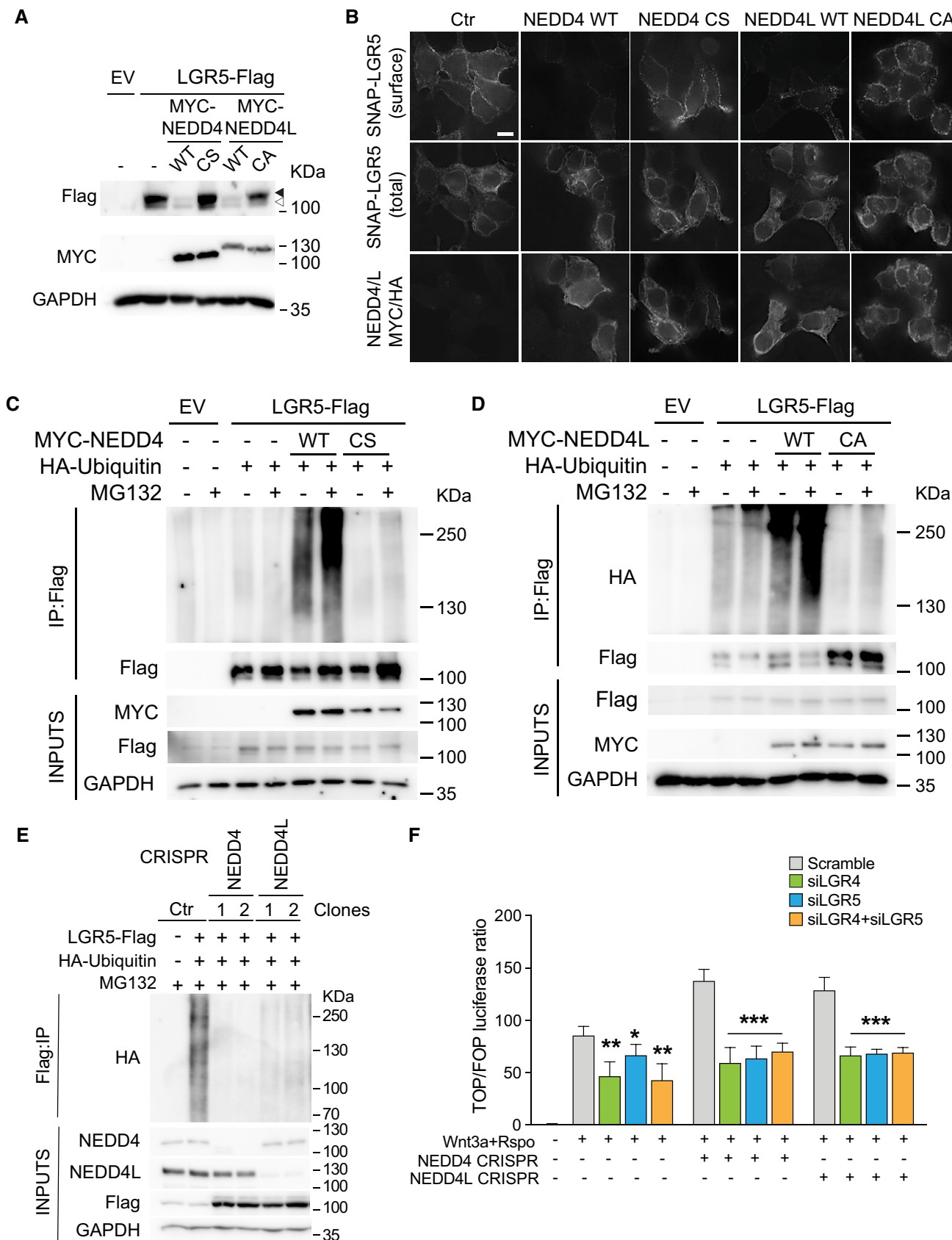

Figure 6.

◀

**Figure 6.  NEDD4 and NEDD4L target LGR5 receptor for lysosomal and proteasomal degradation.**

A    HEK293T cells were transfected with empty vector (EV), LGR5-Flag, MYC-NEDD4 WT or C854S (CS) mutant, MYC-NEDD4L wild-type (WT) or C962A (CA) mutant, followed by Western blot analysis of the indicated antibodies. Black triangle indicates the mature glycosylated form of LGR5, while white triangle indicates the immature unprocessed form of LGR5.

B    Subcellular localisation of SNAP-Lgr5 in the absence or presence of Myc-NEDD4-WT, catalytically inactive HA-NEDD4-CS, Myc-NEDD4L-WT or catalytically inactive HA-NEDD4L-CA. Surface SNAP-Lgr5 was labelled with SNAP Alexa-488 for 10 min. Scale bar, 10 μm.

C, D  HEK293T cells were transfected with constructs expressing LGR5-Flag, HA-Ubiquitin, EV, or the indicated NEDD4 (C) or NEDD4L (D) plasmids. Cells were treated with MG132 followed by anti-Flag IP under denaturing conditions and Western blot analysis using the indicated antibodies.

E    HEK293T control, NEDD4 or NEDD4L CRISPR-mediated mutant cells were transfected with LGR5-Flag and HA-Ubiquitin. Cells were pre-treated with MG132 followed by anti-Flag IP and Western blot analysis using the indicated antibodies.

F    Relative TOPFlash reporter activities of HEK293T cells and NEDD4 and NEDD4L CRISPR cell lines with the indicated siRNA constructs. Cells were treated with Wnt3a supplemented with RSPO-conditioned media. Data represent mean ± standard error of at least three independent experiments. *P*-values were determined using the unpaired two-sided *t*-test (*$P < 0.05$; **$P < 0.01$; ***$P < 0.001$).

## Discussion

The bona fide ISC marker LGR5 is expressed exclusively at the intestinal crypt base under tight regulation, while LGR5[+] ISCs are indispensable for regeneration and tumourigenesis (Metcalfe *et al*, 2014; de Sousa e Melo *et al*, 2017). Transcriptional control of *LGR5* by Wnt, ASCL2 and BMP has been reported in the past (van der Flier *et al*, 2009; Schuijers & Clevers, 2012; Qi *et al*, 2017), yet the regulation of LGR5 protein turnover is largely unknown. To our knowledge, this is the first study describing the post-translational modification of LGR5 receptors via the E3 ubiquitin ligases NEDD4 and NEDD4L.

Ubiquitin modification has been shown to drive cell-surface receptor internalisation and lysosomal degradation (Haglund & Dikic, 2012). For instance, RNF43 and ZNRF3 decrease Wnt signals by ubiquitinating FZD receptors (Hao *et al*, 2012; Koo *et al*, 2012), while deubiquitination has also been shown to play an important role in determining the surface level of FZD by recycling the receptor to the plasma membrane (Mukai *et al*, 2010). Indeed, NEDD4 has been previously shown to target a number of growth factor receptors for internalisation and degradation (Katz *et al*, 2002; Murdaca *et al*, 2004; Persaud *et al*, 2011; Huang *et al*, 2015). The current study provides compelling evidence that NEDD4 and NEDD4L function as Wnt negative regulators by targeting LGR5 receptor for lysosomal and proteasomal degradation (Fig 7D). We demonstrate that both NEDD4 and NEDD4L selectively degrade DVL2 but not DVL1/3. More importantly, we uncovered a new DVL-independent Wnt regulatory role of NEDD4 and NEDD4L by mediating the RSPO receptor LGR5 degradation. Our results further show that loss of Nedd4 and Nedd4l increases ISC numbers and sensitivity to RSPO stimulation, implicating that NEDD4/NEDD4L may contribute to the ISC priming (Yan *et al*, 2017b) by targeting LGR5 for degradation. RSPO potentiates Wnt signalling by forming a complex with LGR4/5 and RNF43/ZNRF3 to neutralise the RNF43/ZNRF3-mediated FZD degradation (de Lau *et al*, 2014). It will be interesting to further investigate the potential overlapping or distinct roles between RNF43/ZNRF3 and NEDD4/NEDD4L in Wnt activation and tumourigenesis. The selective regulatory role of NEDD4/NEDD4L to LGR over FZD receptors further suggests two distinct negative regulatory mechanisms of Wnt and RSPO signalling: ZNRF3/RNF43 for targeting FZD receptors and NEDD4/4L for targeting LGR receptors.

The role of NEDD4 and NEDD4L in cancer progression has been controversial. NEDD4 has been reported as both tumour suppressor and oncogene (Kim *et al*, 2008; March *et al*, 2011; Eide *et al*, 2013; Zeng *et al*, 2014; Lu *et al*, 2016), while the role of NEDD4L in cancer is largely unknown. Our current study provides the first comprehensive analysis of both Nedd4 and Nedd4l in intestinal tumourigenesis. Our data demonstrate that both Nedd4 and Nedd4l exacerbate Apc[min] tumour phenotype, indicating that they are both tumour suppressors. Unlike the previous report showing that Nedd4-deficient tumour phenotype requires surrounding microenvironment (Lu *et al*, 2016), our results show that deletion of Nedd4/Nedd4l in the intestinal epithelium alone is sufficient to enhance Apc[min] tumour growth with significant increase in ISC numbers.

Previous studies have identified several crypt-expressing Wnt inhibitors such as AXIN2, RNF43 and SH3BP4 that are involved in CRC development by targeting Wnt signalling pathway at different subcellular levels (Cancer Genome Atlas, 2012; Giannakis *et al*, 2014, Yan *et al*, 2017a; Antas *et al*, 2019). In this study, we uncover a new crypt-expressing Wnt regulator NEDD4 that targets two Wnt pathway components LGR4/5 and DVL2 to maintain ISC homeostasis and suppress intestinal tumourigenesis (Fig 7D). These findings highlight the complexity and importance of multi-level regulation of the pathway to fine-tune the Wnt signal strength at the crypt and to determine the cellular responses. It is interesting to note that deletion of Nedd4 and Nedd4l was able to exacerbate Apc[min] phenotype despite the observation that NEDD4/NEDD4L inhibit Wnt signalling upstream of APC. The data imply that upregulation of Rspo-Lgr5 signalling upon Nedd4/Nedd4l loss increases tumour predisposition and progression in Apc[min] animals by promoting Wnt activation and ISC self-renewal (Yan *et al*, 2017b). On the other hand, given that NEDD4/4L target both LGR receptors and DVL2 for Wnt regulation, the Wnt activation and stem cell increase in the Nedd4/Nedd4l-deficient intestine is likely caused by both Lgr5 and Dvl2 stabilisation. Despite the clear evidence showing the Lgr5-dependent role in the siRNA assay and the enhanced sensitive to RSPO stimulation *in vivo*, it is admittedly difficult to dissect whether the enhanced stem cell and tumour phenotypes in the mutant intestine are caused by Lgr5 or Dvl2 or both.

In conclusion, our work shows that NEDD4 and NEDD4L are crypt-expressing tumour suppressors by targeting the RSPO receptor LGR5 and DVL2 for degradation. Interestingly, inactivating mutations and deletions of *NEDD4* and *NEDD4L* have also been reported in human CRCs (Cancer Genome Atlas N, 2012) (Appendix Fig S2C), while low NEDD4L expression has been associated with poor prognosis (Tanksley *et al*, 2013). Indeed, expression of *NEDD4L* was profoundly downregulated in CRCs (Appendix Fig S2D),

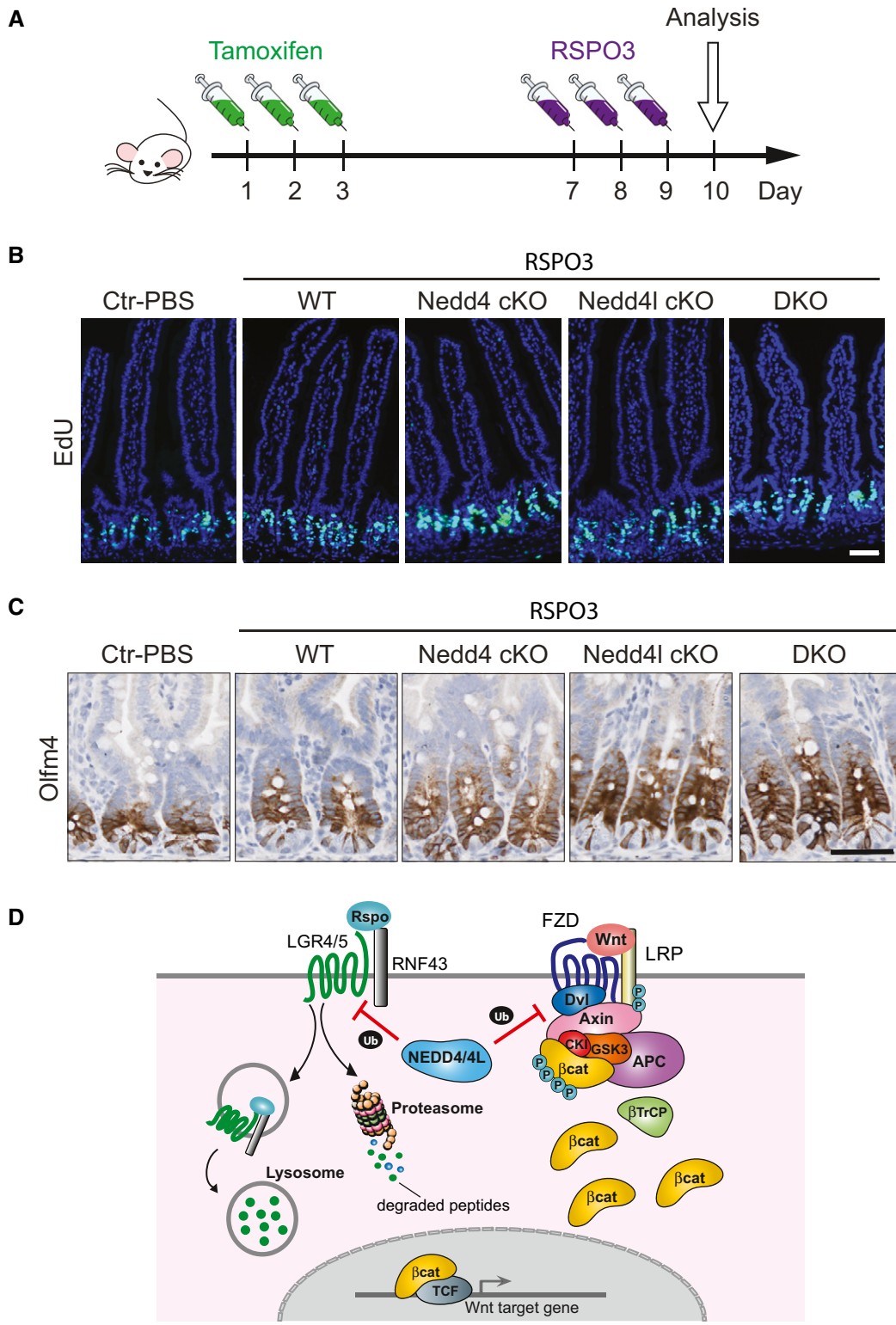

**Figure 7. Loss of Nedd4 and Nedd4l enhances ISC expansion upon RSPO stimulation.**

A    Experimental design of RSPO3 stimulation model.

B, C    Representative images of EdU (B) and Olfm4 (C) staining in intestinal tissues obtained from the indicated genotypes. Scale bar, 50 μm.

D    Model of proposed mechanism of NEDD4/NEDD4L-mediated regulation of Wnt pathway. NEDD4 and NEDD4L target two Wnt pathway components: (i) LGR4/5 receptor for lysosomal and proteasomal degradation, and (ii) DVL2 for proteasomal degradation.

suggesting that inhibition of the tumour suppressors NEDD4/NEDD4L may be an alternative Wnt activating mechanism for some CRCs. Since NEDD4 is expressed in the crypt/stem cell region, it may imply a new negative feedback regulation of Wnt signalling pathway by degrading LGR5 and DVL2 to control the Wnt signal strength in the intestinal crypt.

# Materials and Methods

## Antibodies

β-catenin (610154, BD), caspase-3 (9661L, Cell Signalling), Cyclin D1 (2978S, Cell Signalling), DVL2 (10B5) (sc-8026, Santa Cruz), GAPDH (sc-47724, Santa Cruz), Flag (F3165, SIGMA), HA (Y-11) (sc-7392, Santa Cruz), LGR4 (C-12) (sc-390630, Santa Cruz), Lysozyme (A0099, Dako), c-Myc (9E10) (sc-40, Santa Cruz), NEDD4 (sc-25508, Santa Cruz), NEDD4L (4013S, Cell Signalling), SNAP (P9310S, NEB), Sox9 (AB5335, Millipore) and V5 (ab27671) were used in immunohistochemistry, immunoprecipitations or Western blot analysis.

## Plasmids and other reagents

Full-length NEDD4 was a kindly gift from Igor V. Korobko (Kalinichenko et al, 2012), and NEDD4L was a kindly gift from Olivier Staub (Oberfeld et al, 2011). Full-length human NEDD4, NEDD4L and LGR4 and LGR5 were cloned into XhoI site of pcDNA-Flag, pcDNA-HA or pcDNA-MYC vectors. Various mutants were generated by PCR into pcDNA-MYC or pcDNA-HA. Mutants include NEDD4-CS (C854S) and NEDD4L-CA (C962A). pRK5-HA-ubiquitin-WT was a gift from Ted Dawson (Addgene plasmid #17608). Cycloheximide (C7698, SIGMA) was resuspended in ethanol and used at 50 μg/μl. MG132 (133407-82-6, Calbiochem) was resuspended in DMSO and used at 10 μM for 4 h. Bafilomycin A1 (196000, SIGMA) was resuspended in DMSO and used at 10 nM overnight.

## Cell culture, transfection and TOPFlash assay

HEK293T, DLD1 and HCT116 cells were maintained in DMEM GlutaMAX (Gibco) supplemented with 5% foetal bovine serum (FBS) (Gibco) and 100 units/ml penicillin (Gibco) and 100 μg/ml streptomycin (Gibco). All cell lines were incubated in a humidified atmosphere of 5% $CO_2$ at 37°C. Cells were seeded in plates 24 h before transfection, and plasmids were transfected using Polyethylenimine (Polysciences) according to the manufacturer's instructions. For the TOPFlash luciferase assay, cells in a 48-well plate were transfected with 100 ng of the reporter plasmid TOP or FOP and 10 ng of TK/Renilla for each well. 100 ng of additional plasmid was co-transfected in each well when indicated. After 16 h of transfection, when indicated, cells were treated with Wnt3a-conditioned medium, Wnt3a supplemented with R-spondin or control medium for an additional 18 h. The luciferase activity was measured using a luminometer. For transfections (immunoprecipitation or overexpression), cells were also seeded 24 h before transfection in a 6-well plate and 1–2 μg of plasmid was transfected in each well dependent on each plasmid. Forty-eight hours after transfection, cells were lysed. For siRNA experiments, cells were seeded in 6-well plates 24 h before transfection. The following day, media were changed to antibiotic-free and FBS-free medium and siRNAs (SR310714 and SR322442, OriGene) were transfected using Lipofectamine 2000 (Invitrogen) at 20 nM final concentration. After 5–7 h, transfection mixture was removed and replaced with normal growth medium. 24 h after siRNA transfection, cells were split in a 48-well plate and the TOPFlash luciferase assay protocol was followed.

## Gene editing with CRISPR/Cas9 system

To generate NEDD4 and NEDD4L mutants, HEK293T cells were transfected with plasmids encoding Cas9 (41815, Addgene) modified to add puromycin resistance and guideRNAs (gRNAs) (gRNA-GFT-T1 was, #41819, Addgene) as previously described (Novellasdemunt et al, 2017). For NEDD4 targeting, gRNA1, 5′-AAGTCCGGCATGCAC-CAAAT-3′ or gRNA2, 5′-GACTCTTACCGGAGAATTAT-3′ was used. For NEDD4L targeting, gRNA1, 5′-TATGGACTTTCCGAAGACGA-3′ or gRNA2, 5′-AAGAGATGTTCAACCCCTAC-3′ was used.

To generate DVL2 CRISPR mutants, HEK293T cells were transfected with plasmids encoding Cas9 (pSpCas9(BB)-2A-Puro (PX459) V2.0) in which we introduced our guideRNA (gRNA). This plasmid was a gift from Feng Zhang. For DVL2, gRNA1, 5′-GACGAAGGT-GATTTACCACC-3′ was used. The gRNAs were targeting specific genomic loci. We screened the potential targeted cells by immunoblotting and then confirmed by genomic DNA sequencing.

## SNAP-surface immunofluorescence

HEK293T cells were grown on laminin-coated glass coverslips. Cells were co-transfected with 100 ng SNAP-LGR5 or SNAP-FZD5 and 150 ng myc-NEDD4, HA-NEDD4-CS, myc-NEDD4L, HA-NEDD4L-CA or pcDNA4 as a control with Fugene according to the manufacturer's instructions. After 24 h of transfection, cells were labelled with 1 μM SNAP-Surface Alexa 488 (NEB) for 15 min at 4°C to block endocytosis. Cells were then immediately washed with fresh RPMI media and fixed in 4% paraformaldehyde. Cells were incubated with primary antibodies NEDD4 (Santa Cruz) or NEDD4L (Cell Signaling) for 1 h at RT followed by a secondary antibody conjugated to Alexa-568 (Invitrogen) and DAPI (Sigma) in 2% BSA-PBS (Roche). Cells were mounted in Prolong Diamond (Life technologies) and imaged using a DeltaVision Core system.

## Real-time quantitative RT–PCR

RNA was extracted according to the manufacturer's instructions (Qiagen RNAeasy). cDNA was prepared using Maxima first strand cDNA synthesis (#1672, Thermo Scientific). Quantitative PCR detection was performed using iTaq SYBR Green Supermix (#172-5121, Bio Rad) using specific primers to: mAscl2: F: 5′ AATGCAAGCTT-GATGGACGG 4′ R: 5′ GGAAGCCCAAGTTTACCAGC 3′; mAxin2: F: 5′TCCAGAGAGAGATGCATCGC 3′ R: 5′ AGCCGCTCCTCCAGACTATG 3′; mCD44: F: 5′ GGCTCATCATCTTGGCATCT 3′ R: 5′ GCTTTTTCTT CTGCCCACAC 3′; mHprt1: F: 5′TCATGAAGGAGATGGGAGGC 3′ 5′ CCACCAATAACTTTTATGTCCCC 3′; mLgr5: F: 5′ CATCAGGTCAA-TACCGGAGC 3′ R: 5′ TAATGTGCGAGGCACCATTC 3′; mNedd4: F: 5′ GTGCAGACTCACCTTGCAGA 3′ R: 5′ TTTTTCTTCCCAACCTGGT G 3′; mNedd4l: F: 5′ CAACTTGGACTCGGCCAATC 3′ R: 5′ GTTACTG

TTGGCGAGCTGAG 3′. After cDNA amplification (40 cycles), samples were normalised to Hrpt1 and data were expressed as mean ± SD.

### IP and immunoblotting

When indicated, cells were pre-treated with 10 nM Bafilomycin A1 overnight or 10 μM MG132 proteasome inhibitor for 4 h prior to lysate collection. Cells were washed and collected with cold PBS and lysed in cold lysis buffer containing 150 mM NaCl, 30 mM Tris (pH 7.5), 1 mM EDTA, 1% Triton X-100, 10% glycerol, 0.5 mM DTT, protease and phosphatase inhibitor cocktail (Thermo Scientific, 78446). After clarification by centrifugation (18,800 $g$ for 30 min at 4°C), the cellular lysates were precleared with IgG-agarose beads (Millipore, 16-266) for 1 h at 4°C and the supernatants were immunoprecipitated with the indicated antibodies or anti-Flag-M2 affinity beads (A220, Sigma) at 4°C overnight. Immunocomplexes were washed with cold lysis buffer six times, resuspended in SDS sample buffer, and subjected to SDS–PAGE and Western blot analysis.

### IP under denaturing conditions

When indicated, cells were treated with 10 μM MG132 proteasome inhibitor for 4 h prior to lysate collection. Cells were collected in denaturing lysis buffer (1% SDS, 5 mM EDTA, 10 mM DTT, 5 mM NEM, protease and phosphatase inhibitor cocktail (Thermo Scientific, 78446)). Then, cells were mixed by vortexing for 2–3 s at maximum speed and the samples were boiled at 95°C for 5 min to denature. The cells suspension was then diluted 1/10 in the IP lysis buffer, and the lysate was passed through a needle several times to fragment the DNA followed by immunoprecipitation protocol as described before.

### Immunohistochemistry and Edu staining

For analysis of small intestine and colon by immunohistochemistry and Edu staining, tissues were fixed in 10% formalin and embedded in paraffin. For small intestinal tissues, same proximal part of the small intestine from all genotypes was used throughout the study. Immunohistochemistry was performed as described (Novellasdemunt et al, 2017). The buffers used for antigen retrieval were citrate (Sox9 and Caspase-3) or Tris-EDTA (ß-catenin and lysozyme). Edu staining was performed following manufacturer's instructions (C10338, Invitrogen).

### RNAScope *in situ* hybridisation

*In situ* hybridisation (ISH) for Lgr5, Olfm4, Nedd4 and Nedd4l was performed using the RNAScope FFPE assay kit (Advanced Cell Diagnostics) according to the manufacturer's instructions. Briefly, 4 μm formalin-fixed, paraffin-embedded tissue sections were pre-treated with heat and protease digestion before hybridisation with the target probe. Then, an HRP-based signal amplification system was hybridised to the target probes (Lgr5, 312171; Olfm4, 3111831) before colour development with 3,30-diaminobenzidine tetrahydrochloride (DAB). 20 crypts from 3 mice per group were used to quantitate the number of Olfm4$^+$ cells.

### Intestinal organoid culture

Organoids were established from freshly isolated wild-type, Nedd4 cKO, Nedd4l cKO and DKO small intestine. Tissues were incubated in cold PBS containing 2 mM EDTA for isolating epithelial crypts and culture as previously describe (Sato et al, 2009) except that Matrigel was replaced with Cultrex® BME, Type 2 RGF PathClear (Amsbio 3533-010-02). In brief, the organoid basal media contain EGF (Invitrogen PMG8043), Noggin and R-spondin (ENR) (5%). For the R-spondin withdrawal experiment, when indicated, R-spondin was used at 1%. Noggin and R-spondin-conditioned media (CM) were generated from HEK293T cells. For all the other organoid experiments, Intesticult™ Organoid Growth media (06005, Stem Cell Technologies) was used. The Rho kinase inhibitor Y-27632 (Sigma) was added to the culture when trypsinised.

### Edu staining in organoids

Organoids were grown in 15 μl of RGF BME into an 8-well chamber side (Lab-Tek II, 154534). When indicated, 10 μM Edu was added to the growth media for 2 h before fixing. Edu staining was performed as described (Novellasdemunt et al, 2017).

### Organoid colony formation assay

Organoids were trypsinised and counted. 2,000 single cells were seeded in BME per 48 wells and placed in a 37°C incubator to polymerise for 20 min. 250 μl of IntestiCult™ Organoid Growth media plus Y-27632 was then added and cultured for 5 days. Number of spheres formed in each well was counted as plating efficiency. Experiments were performed in triplicate.

### Generation of Lgr5-HA knock-in mouse intestinal organoids

The knock-in of Lgr5-HA tag was performed by electroporation using an oligo donor (TTACCCCATGACTGAAAGCTGTCATCTCTCT TCAGTTGCATTTGTCCCATGTCTCTACCCATACGATGTTCCAGATT ACGCTGCGGCCGCATAGTGACTATGAGAGAGGAACGTTTTTAAGC GTTTGAAACCTGAAAAGTGATTTCTATCAGAGCAGTAGCTAAGAA AAGCTGAG) together with a plasmid encoding Cas9 (pSpCas9(BB)-2A-Puro (PX459) V2.0) in both WT and DKO organoids. Three days after the electroporation, puromycin (1.5 μg/ml) was added for 3-day selection. Organoids were then passaged once for expansion and were collected for protein extraction and Western blot analysis. This plasmid was a gift from Feng Zhang. The gRNA was targeting C-terminal region of mouse Lgr5 locus. We screened the potential targeted cells by immunoblotting. Electroporation was performed following manufacturer's instructions (VPI-1005, Lonza).

### Animal procedures

All animal regulated procedures were carried out according to Project License constraints (PEF3478B3 and 70/8560) and Home Office guidelines and regulations. In accordance with the 3Rs, the smallest sample size was chosen that could show a significant difference. Nedd4$^{fl/fl}$ and Nedd4l$^{fl/fl}$ mice were obtained from Max-Planck-Institute of Experimental Medicine, Goettingen, Germany, Department of Molecular Neurobiology. Nedd4$^{fl/fl}$ mouse was

generated as detailed (Kawabe *et al*, 2010). Nedd4l$^{fl/fl}$ mouse was generated as explained (Shi *et al*, 2008). Animals of both sexes at age 6–7 weeks on C57/BL6J background were used for the different experimental conditions and harvested as indicated.

Tamoxifen was injected intraperitoneally for 3 consecutive days (1.5 mg/10 g of mouse weight) from a 20 mg/ml stock solution. 5-ethynyl-20 deoxyuridine (EdU) (Life Technologies) was injected intraperitoneally (0.3 mg/10 g of mouse weight) from a 10 mg/ml stock solution.

For the RSPO3 experiments, mice were injected with tamoxifen as detailed above, and 7 days later, RSPO3 (10 mg/kg in PBS) was injected intraperitoneally for three consecutive days. Mice were culled 24 h later, and the tissue was collected to be analysed.

### Quantification and statistical analysis

Statistical analyses were performed using GraphPad Prism 8 software. Statistical details and sample numbers are specified in the figure legends. For parametric data, statistical significance was determined using student's unpaired, two-tailed *t*-test. For survival experiments, Log-rank (Mantel–Cox) test was used. *P* values are represented as *$P < 0.05$; **$P < 0.01$; ***$P < 0.001$.

**Expanded View** for this article is available online.

### Acknowledgements
We thank H. Kawabe for providing the Nedd4-floxed and Nedd4l-floxed animals and Andrés Méndez Lucas for assistance with animal experiments. We thank the Francis Crick Institute's Experimental Histopathology, Flow Cytometry and Biological Research Facilities. This work was supported by the Francis Crick Institute, which receives its core funding from Cancer Research UK (CRUK) (FC001105), the UK Medical Research Council (FC001105) and the Wellcome Trust (FC001105). Work in the V.S.W.L laboratory was also supported by the European Union's Horizon 2020 research and innovation programme (668294). The work of M.M.M. is part of the Oncode Institute, which is partly financed by the Dutch Cancer Society, and is supported by the Netherlands Organization for Scientific Research (NWO VICI Grant 91815604; ZonMW-TOP grant 91218050).

### Author contributions
VSWL conceived the project. VSWL and LN designed the experiments and analysed data. LN, AK, CJ, MP-B, AB and PA performed the experiments. CJ and MMM designed and analysed SNAP-LGR5 experiment. JV and HG generated RSPO3 protein. MMM provided scientific advice on LGR5 degradation data. VSWL and LN wrote the manuscript, which was reviewed by all authors.

### Conflict of interest
The authors declare that they have no conflict of interest.

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
