## [Review Process File · The EMBO Journal]

NEDD4 and NEDD4L regulate Wnt signalling and intestinal stem cell priming by degrading LGR5 receptor

Laura Novellasedemunt, Anna Kucharska, Cara Jamieson, Maria Prange-Barczynska, Anna Baulies, Pedro Antas, Jelte van der Vaart, Helmuth Gehart, Madelon M. Maurice, Vivian S. W. Li

Review timeline:	Submission date:	24th Jun 2019
	Editorial correspondence:	16th Jul 2019
	Author correspondence:	18th Jul 2019
	Editorial Decision:	28th Jul 2019
	Revision received:	14th Oct 2019
	Editorial Decision:	8th Nov 2019
	Revision received:	25th Nov 2019
	Accepted:	28th Nov 2019

Editor: Daniel Klimmeck

Transaction Report:

Editorial correspondence 16th Jul 2019

Thank you again for the submission of your manuscript (EMBOJ-2019-102771) to The EMBO Journal. Your manuscript has been assessed by two referees whose comments are enclosed below. In light of their input together with our editorial considerations in the team, I am afraid we decided that we cannot offer publication in The EMBO Journal.

As you will see, the referees acknowledge the potential interest of your work, although they also express major concerns. In particular, referee #1 raises major issues regarding the novelty of your findings in light of earlier *in vivo* data showing a tumor suppressor function for Nedd4 in CRC as well as linking Nedd4 to Wnt signaling. Further, this referee is not convinced that the role of Nedd4-L in Lgr5 turnover *in vivo* and ISC proliferation is conclusively demonstrated, which in his/her view undermines the impact of the current results. Further, this reviewer. Referee #2 states that the pathophysiological relevance of your findings remains unclear and agrees with referee #1 in that causalities between Nedd4-L deficiency-evoked phenotypes and Lgr5 regulation are not sufficiently supported by the data, also pointing to ambiguities with Dvl2-dependent roles of Nedd4. Finally, the referees raise issues related to missing controls, data and methods representation as well as inconsistencies between data or towards literature that impact the level of robustness provided.

These are important points at the core of the proposed advance in our view, and taking into account the extensive revisional work required to address all these matters and the one major round of revisions we offer at the EMBO journal, I am afraid we have concluded that we cannot offer to publish your work here.

REFeree REPORTS:

Referee #1:

In this manuscript, the authors studied the role of Nedd4-1 (Nedd4) and Nedd4-2 (Nedd4L) in WNT signaling in the intestine and in colon cancer, and the role of these E3 ligases in targeting LGR5 (and Dvl2) in the intestinal stem cells. They show that these Nedd4s suppress WNT signaling and suggest that this suppression is mediated by targeting LRG5 for degradation.

Evaluation:

1. This MS has some interesting findings, but overall it is not particularly novel. The finding that Nedd4 suppresses WNT signaling in the colon and that intestinal Nedd4 knockout (KO) enhances APC(min)-driven colonic tumor growth was already published several years ago (Lu et al, *Oncogene* 2016- cited in the current MS), as was the fact that Dvl is a substrate for Nedd4L (Ding et al. *JBC* 2013- also cited here). The new finding in the current MS is the fact that the Nedd4 close relative Nedd4L also contributes to the regulation of colonic tumor growth (ie has redundant function with Nedd4 in this context, a finding that may be interesting, but not earth-shaking), and that LGR5 may be a target for Nedd4/Nedd4L regulation. The latter observation is potentially important, but the authors have not provided any in vivo (in animal) proof for it, only in transfected cells, as detailed below.
2. Proof that LRG5 is a substrate of Nedd4/Nedd4L in vivo is missing:
To correct this, the authors need to show: (i) the accumulation of LRG5 protein in the stem cells of the small intestine and colon of the Nedd4/Nedd4L KO or DKO mice. (ii) that Nedd4/Nedd4L bind LRG5 (including Co-IP of endogenous proteins), if indeed LRG5 is a substrate for Nedd4/Nedd4L; (iii) test Nedd4/Nedd4L-mediated ubiquitination of LGR5 using a proper ubiquitination assay: the ones shown in the paper were not done correctly, since samples have to be boiled in SDS prior to the LRG5 IP and immunoblotting for ubiquitin (to ensure that the ubiquitination observed is of LRG5 itself and not its associated proteins); (iv) all pulse-chase assays to analyze LRG5 stability upon loss of Nedd4 or Nedd4L need to be repeated several times and quantified. The IF studies (Fig 6 & Suppl. 5D) need to be quantified as well.
3. For all intestinal sections immunostained for different markers (eg Fig 1): The sections appear to come from different parts of the small intestine (and from different mice?), so that may affect the results (ie pattern of expression of the same marker may differ between sub-sections of the intestine, such that comparison between expression of the different markers may not be meaningful, as it may represent different parts of the intestine rather than differences between WT vs KO mice). I.e. the real way to perform such comparison of expression pattern of different markers is to use serial sections, and to indicate exactly where they came from, which should be indicated in the legend (it is not enough to just write: intestinal crypt). Also, were siblings used for the comparisons in the study?
4. What is the genetic background for all mice and crosses used in the study? This is very important both for the current study, and for comparison with earlier published studies.
5. Fig 2C (organoids): why is the Edu staining (cell proliferation) lower in the Nedd4-KO or DKO mice relative to WT? This contradicts the statements in the paper. Also, Suppl. Fig 2B (organoids) should be quantified.

Referee #2:

In this manuscript, Novellademunt et al investigated the role of the E3 ligase Nedd4/Nedd4L in normal intestinal crypt homeostasis and colon cancer driven by APCmin. They find that Nedd4/Nedd4L KO in the intestinal epithelium induces the proliferation of intestinal crypts cells and the number of ISCs. They claim that Wnt signaling is elevated in DKO organoids, which undergo reduced apoptosis upon limited addition of Rspodin. They also show that Nedd4/Nedd4L exacerbates intestinal tumor grade and phenotype by APCmin background. Using 293T cell lines to delineate the mechanism, they show that Nedd4/Nedd4L induces ubiquitination of LGR5, leading to proteasomal and lysosomal degradation as well as degradation of Dvl2, thereby reducing Wnt signaling. The manuscript is well written and the findings are novel (E3 ligase for LGR5) and are well supported. However, there are some critical points need to be addressed to strengthen their study.

Major points

1. A main concern is whether the in vivo effects from DKO can solely be explained by LGR5 regulation. Since the authors also observed Dvl2 degradation by Nedd4/Nedd4l, increased LGR5 or increased Dvl2 or both can cause hyperproliferation of the crypt cells in DKO. To prove that Nedd4/Nedd4l KO increases the proliferation and tumor phenotype via LGR5 regulation, some rescue experiments are needed. For example, if they can rescue the survival rate of APCmin Nedd4/Nedd4l DKO by crossing with LGR5^{+/-}, this would support their hypothesis and also rule out the effect from Dvl2 regulation. However, such a time consuming experiment may be beyond the scope of this study. In its absence, the claim that in vivo effects from DKO are solely via LGR5 regulation should be toned down.
2. In Figure 1, the author only show Olfm4 as ISC marker. Multiple ISC markers should be checked such as LRIG1, Ascl2, LGR5.
3. The authors mainly rely on overexpression to prove their mechanism (LGR4/5, Dvl2). Can they try to detect endogenous LGR4 or LGR5 and Dvl2 to show the effect of Nedd4/Nedd4l overexpression or knockout?
4. The evidence that Nedd4/Nedd4l regulates Wnt signaling only via LGR5, is weak. To further prove their point Nedd4/Nedd4l lof effect in Hek293 cells on Topflash should be compared between Wnt3a only and Wnt3+Rspo. Furthermore, Nedd4/Nedd4l lof Topflash activity should be compared {plus minus} siLGR4,5.
5. For the ubiquitination assay in Figure 6, the author show that Nedd4/Nedd4l can ubiquitylate LGR4/5 by pull down of LGR4/5 and detecting HA-UB in cell lysates. To provide more direct evidence for this model, an in vitro ubiquitination assay should to be performed with recombinant proteins. At the minimum, pulled-down sample could be boiled with 1% SDS to remove LGR4/5 bound ubiquitinated proteins; dilute 10 times with lysis buffer and pull down it again with the same antibody to detect ubiquitination of LGR4/5. This way, the authors can at least conclude that ubiquitination is not due to other proteins binding to LGR4/5.

Minor points

1. From RNA scope results, Nedd4 expression is more restricted in the crypts than Nedd4l. However, in many figures, the effect of Nedd4 KO is milder than Nedd4l KO (Figure 2A, C, D, Figure 3C, D, Figure 4A and Supplement 3A and B). Have the author seen the increased Nedd4l level in Nedd4 KO as a compensation? Or is there an explanation for this?
2. Comparing Figure 6A and B for LGR5 level, Nedd4 completely removed the LGR5 band from the western blot while LGR5 level seems not be changed upon Nedd4 overexpression from the immunofluorescence picture. If the authors took the picture where there are still significant LGR5 signaling in the cells, it should be mentioned in the text or legends.
3. Figure legend for Figure 6B is not matched with Figure 6B. Please check the Figure legend in the corresponding supplement figure.
4. Nedd4/Nedd4l overexpression in 293T cell or using Nedd4 or Nedd4l KO 293T cell lines, the authors need to prove mRNA levels of LGR4, 5 are not changed.
5. In Figure 5 E-G, the authors showed that Nedd4/Nedd4l overexpression decreased Topflash in HCT116 but not in APC4 and DLD1. However, there is a study (PMID : 23349017; Al-Kharusi at al, 2013, Carcinogenesis) showing that there is no detectable LGR5 protein in HCT116. Thus, it is worth to show that LGR5 is expressed in HCT116, APC4, DLD-1 (qPCR or western blot).

Thank you very much for the timely review. We appreciate very much your effort and the efficiency of the review process (the fastest record I have ever had in the past!), and your suggestion for the transfer.

While the reviewer's comments are relevant and will certainly help improve and strengthen our manuscript, we feel that some of the comments/suggestions from reviewer #1 are not justified or reasonable. First of all, reviewer #2 provided very reasonable advice and comments and we will be able to fully address. For reviewer #1, we will also be able to address most of the concerns and suggestions. There are two major concerns that I believe are the reasons that you recommended against publication in EMBO J. First is the novelty, and second is the *in vivo* proof of Lgr5 as the target.

Regarding the novelty, what was published before was Nedd4l inhibits Wnt by targeting Dvl2, and Nedd4 increases adenoma growth in full-body KO with poor characterisation. Our current findings uncover the novel role of Nedd4 by targeting Lgr5 receptor, a key stem cell priming factor for degradation. This part was recognised by both reviewer as of importance to the field. I also need to emphasize that the previously reported Nedd4 KO mouse data (Lu et al Oncogene 2016) was poorly characterised and could not be reproduced. It is important to publish a comprehensive analysis of the *in vivo* data with proper stem cell and tumour analysis to show the importance of both Nedd4 and Nedd4l in intestinal stem cell and tumour development.

Regarding the *in vivo* proof part, we feel that this is not a reasonable comment. It is well recognised in the field that there is no reliable endogenous antibodies recognising Lgr5 receptor for validation *in vivo*. This is evident by the pioneer group Hans Clevers who published the Lgr5 as ISC marker, he never published any paper using antibody recognising endogenous Lgr5, simply because the existing ones are not reliable. We have been in close contact with the Clevers group, hoping to obtain reliable endogenous Lgr5 antibodies without success. In that case, it is not justified to request *in vivo* validation of protein stabilisation when there is no existing reliable antibody available. In fact, many groups published important findings about post-translational modification of Wnt receptors using similar strategies as us, demonstrating the phenotypes *in vivo* and define the mechanism *in vitro* (Lgr as receptors for Rspodin by de Lau et al Nature 2011; RNF43-mediated Fzd receptor degradation by Koo et al Nature 2012; RAL GTPase-mediated Fzd7 internalisation by Johansson et al Cell Stem Cell 2019). This is a widely accepted approach in the field, thus asking for *in vivo* proof of Lgr5 stabilisation in the absence of endogenous antibody is unreasonable. Importantly, we have provided functional proof by testing the Rspodin response in *ex vivo* organoids and *in vivo* Rspo injection (Figure 7), which is the ultimate proof that Lgr receptors are stabilised *in vivo*. Unfortunately the data was dismissed or under-appreciated by reviewer #1. Thus, we feel that these concerns raised by reviewer #1 was not justified.

On the other hand, we could address these questions using *ex vivo* organoids. We could generate Lgr5-KI-tag to proof the protein stabilisation. Also, questions 3-5 by reviewer #1 are all based on misunderstanding, which we can address easily. Based on the reasons above, I'd like to kindly ask for reconsideration of our manuscript in EMBO journal. We will be happy and ready to address the rest of the reviewers' concern for a timely resubmission to EMBO J. Thank you for considering it and please let me know if you have any questions.

Thank you for contacting me regarding our decision and for your patience with my response, which got delayed due to internal discussions in the team regarding your letter together with re-assessing the manuscript and referee comments.

After looking back into literature mentioned, we appreciate your point regarding the difficulty to employ an antibody on LGR5 endogenous. In addition we realise that you would - judging from the information provided in the your letter - be potentially able to address the issues raised by the referees in a revised version of the manuscript. Overall, we would thus invite you to work towards a

re-review. Accordingly, we would - given that you addressed all the experimental issues with compelling data - be prepared to ask the referees for further input.

Please note however, that in particular the proof for an in vivo relevant functional link between Nedd4-L and Lgr5 as well as the distinction between Lgr5 versus Dvl2-dependent activities of Nedd4 has been major concerns of the referees, thus will in our view be core aspects to be considered.

Given the importance and broad interest of the question addressed, it would be essential for you to provide a definitive and accurately described dataset in the revised version.

Please contact me if you have any questions, need further input on the referee comments or if you consider engaging in a compelling revision, in which case we would not close the file.

However, please note, that since the results of your experiments are entirely open at this stage, we cannot in any way predict the outcome of a re-submission, or make any promises towards publication.

Revision - authors' response

14th Oct 2019

Referee #1:

In this manuscript, the authors studied the role of Nedd4-1 (Nedd4) and Nedd4-2 (Nedd4L) in WNT signaling in the intestine and in colon cancer, and the role of these E3 ligases in targeting LGR5 (and Dvl2) in the intestinal stem cells. They show that these Nedd4s suppress WNT signaling and suggest that this suppression is mediated by targeting LRG5 for degradation.

Evaluation:

1. This MS has some interesting findings, but overall it is not particularly novel. The finding that Nedd4 suppresses WNT signaling in the colon and that intestinal Nedd4 knockout (KO) enhances APC(min)-driven colonic tumor growth was already published several years ago (Lu et al, Oncogene 2016- cited in the current MS), as was the fact that Dvl is a substrate for Nedd4L (Ding et al. JBC 2013- also cited here). The new finding in the current MS is the fact that the Nedd4 close relative Nedd4L also contributes to the regulation of colonic tumor growth (ie has redundant function with Nedd4 in this context, a finding that may be interesting, but not earth-shaking), and that LGR5 may be a target for Nedd4/Nedd4L regulation. The latter observation is potentially important, but the authors have not provided any in vivo (in animal) proof for it, only in transfected cells, as detailed below.

Response:

We are grateful to the reviewer's comment, and we appreciate the reviewer's concerns and constructive suggestions. These are all relevant to the conclusion of the study, and have now been addressed accordingly in the specific comments below.

2. Proof that LRG5 is a substrate of Nedd4/Nedd4L in vivo is missing:

To correct this, the authors need to show: (i) the accumulation of LRG5 protein in the stem cells of the small intestine and colon of the Nedd4/Nedd4L KO or DKO mice.

Response:

We appreciate the reviewer's concern and suggestion about the in vivo proof of Lgr4/5 as a substrate of Nedd4/Nedd4L. Although this is a fair suggestion, it is practically challenging due to the lack of reliable commercially available Lgr5 antibodies. In fact,

there is lack of publication showing endogenous Lgr4/5 protein level in mouse intestine in vivo due to the same reason. Most publications showed Lgr5 mRNA expression either using Gfp KI animals or via RNA in situ hybridization. We have sought advice from the pioneer groups of Lgr5 work, Hans Clevers and Nick Barkers, and they both admitted that there are no reliable Lgr5 antibodies that work in their hands. In our hand, we have tried at least 4 different antibodies and none of them offer reliable results on either immunostaining, Western blotting or FACS sorting. We have even obtained customised antibodies generated from Nick Barker's lab from Singapore (who originally discovered Lgr5 as intestinal stem cell marker) and have tested extensively in our lab. Of note, Nick told us that those two antibodies were still in the testing stage and were unlikely to work in mice. Here are FACS results:

Antibody 1: Origene (TA503316) conjugated with Alexa Fluor 647 or APC

VillinCre Mouse, A647 secondary antibody

VillinCre Mouse, APC secondary antibody

Antibodies 2 and 3: Kind gift from Nick Barker's Lab (D2 and G4), unconjugated

VillinCre Mouse

Antibody 4: R&D PE-conjugated (FAB8240P)

VillinCre organoids

Apc mutant organoids

Overall, none of the antibodies tested can actually detect *Lgr5* protein in either mouse or human samples. The FACS results are the same with or without the primary antibodies, indicating that the signal detected was just non-specific noise. The only antibody that showed increased signal was #4 from R&D (FAB8240P). However, when we further analysed the *Gfp*+ve population from *Lgr5*-*Gfp* intestine, there was no enrichment at all using this antibody, indicating that the signal was not specific to *Lgr5*. This is perhaps not so surprising, given that there is no peer-reviewed publication so far showing endogenous *Lgr5* protein expression in mouse intestine. However, we did show functionally that *Nedd4*/*Nedd4l* KO intestine demonstrated increased sensitivity to *R-spondin* treatment. *R-spondin* is the ligand of *Lgr4/5* receptor, which is a functional proof that *Lgr4/5* receptors are indeed upregulated in the KO animals.

To further attempt to demonstrate *Lgr5* protein accumulation *in vivo*, we have now generated *Lgr5*-HA tagged KI organoids using CRISPR targeting (new Supplementary Fig. 6D-6E).

The results showed that endogenous *Lgr5*-HA tagged protein was indeed stabilised in DKO as compared to WT organoids. Loss of *Nedd4* and *Nedd4l* protein in the DKO organoids was further confirmed by endogenous Western blotting analysis. This is our best attempt to address the accumulation of *Lgr5* protein in the DKO organoids.

(ii) that *Nedd4*/*Nedd4l* bind LRG5 (including Co-IP of endogenous proteins), if indeed LRG5 is a substrate for *Nedd4*/*Nedd4l*;

Response:

We thank the reviewer's suggestion. Unfortunately, as mentioned above, none of the LGR5 antibodies that we have tried gave reliable result for detecting endogenous LGR5 protein level by Western blotting. On the other hand, we have found one antibody recognising endogenous human LGR4 protein (#sc-390630, Santa Cruz). Since *Nedd4*/*Nedd4l* targets both *Lgr4* and *Lgr5*, we decided to perform Co-IP of endogenous proteins LGR4 as suggested by the reviewer using LGR4, *DVL2*, *NEDD4* and *NEDD4L* antibodies (new Supplementary Fig. 6A).

The results showed that NEDD4 and NEDD4L indeed bound to LGR4 and DVL2 endogenously.

(iii) test Nedd4/Nedd4L-mediated ubiquitination of LGR5 using a proper ubiquitination assay: the ones shown in the paper were not done correctly, since samples have to be boiled in SDS prior to the LRG5 IP and immunoblotting for ubiquitin (to ensure that the ubiquitination observed is of LRG5 itself and not its associated proteins);

Response:

We thank the reviewer's constructive suggestion to improve our manuscript. To further ensure that the ubiquitination observed is of LGR5 itself rather than its associated proteins, we have now repeated the experiment in boiled SDS condition as suggested by the reviewer. Consistent with our original data, the new results showed that LGR5 ubiquitination was strongly enhanced upon expression of WT NEDD4 or NEDD4L constructs but not the catalytic inactive mutants (new Fig. 6C-6D).

(iv) all pulse-chase assays to analyze LRG5 stability upon loss of Nedd4 or Nedd4L need to be repeated several times and quantified. The IF studies (Fig 6 & Suppl. 5D) need to be quantified as well.

Response:

We have now quantitated the pulse-chase assays and IF data in Fig. 6 and Supplementary Fig. 5.

Quantitation of CHX pulse-chase data:

Quantitation of SNAP data (new Supplementary Fig. 5E):

3. For all intestinal sections immunostained for different markers (eg Fig 1): The sections appear to come from different parts of the small intestine (and from different mice?), so that may affect the results (ie pattern of expression of the same marker may differ between sub-sections of the intestine, such that comparison between expression of the different markers may not be meaningful, as it may represent different parts of the intestine rather than differences between WT vs KO mice). The real way to perform such comparison of expression pattern of different markers is to use serial sections, and to indicate exactly where they came from, which should be indicated in the legend (it is not enough to just write: intestinal crypt). Also, were siblings used for the comparisons in the study?

Response:

The reviewer was concerned if the immunostaining sections were coming from different parts/regions of the small intestine among different animals. We apologise for the confusion. Indeed, all tissues analysed in the immunostaining in Fig.1 and Suppl Fig.1 (both WT and KO) were from the same proximal regions of the small intestine. There might be slight crypt-villus length variability due to tissue processing, which might give the wrong impression that they were from different regions. Also, the length of the villi was slightly longer in DKO intestine in general, albeit not significant. We

have now replaced some images in Fig.1 with more representative images, and have indicated clearly in the method and legends about the region of the intestine.

Importantly, we have indeed provided quantitation of the staining in Fig. 1 to complement the representative images, which was included in the original Fig. 1O-T. To further clarify the results, quantitation of Edu+ cells were counted in 10 crypts per animals ($n=3$ per group). Crypt length, Olfm4+, Cyclin D1+, Sox9+ and Lysozyme+ cells was counted in 20 crypts per animal per genotype with $n=3$ for each genotype. This means that each dot in the graph represents data generated from average of 20 crypts per animal. This has now been clarified in the figure legend and method.

Regarding the reviewer's last point, WT and DKO animals were not direct siblings. In order to obtain reliable numbers of DKO ($VillinCreERT2;Nedd4^{hom};Nedd4^{hom}$) animals for analysis, $VillinCreERT2;Nedd4^{hom};Nedd4^{het}$ were used as the breeding pairs for DKO. Thus, it was impossible to obtain pure VillinCreERT2 strain in the sibling as WT control in our breeding strategy. Although some researchers used floxed animals without VillinCreERT2 as WT control, recent publication has reported genome toxicity from CreERT2 activation (Bohin N Stem Cell Reports 2018), indicating the importance of using VillinCreERT2 as WT control. Although WT and DKO animals were not direct siblings, we stress that the VillinCreERT2 WT mice used in this study were the original parental lines to generate the DKO animals. Importantly, we chose the animals with same age for proper comparison.

4. What is the genetic background for all mice and crosses used in the study? This is very important both for the current study, and for comparison with earlier published studies.

Response:

All our animals involved in the study were on C57/BL6J background, which is the same as the previously published ones. We apologise for the missing information in the method, which has now been updated.

5. Fig 2C (organoids): why is the Edu staining (cell proliferation) lower in the Nedd4-KO or DKO mice relative to WT? This contradicts the statements in the paper. Also, Suppl. Fig 2B (organoids) should be quantified.

Response:

The reviewer raised concern about the Edu staining in Fig. 2C. The images chosen in the figures were representative pictures for the individual genotype. We have now provided more images below to give a better idea of the Edu staining patterns for each genotype.

Overall, it is obvious that Edu staining in WT organoids were mostly restricted to the budding crypts, which is consistent with the data reported by many other groups. On the other hand, the number of Edu+ cells were clearly increased in all three mutants as compared to WT, and the staining has extended to the lumen rather than crypt-exclusive. Similar to *in vivo* data, the proliferation increase was more striking in Nedd4I cKO and DKO as compared to Nedd4 cKO. However, to be more quantitative, we have performed organoid formation assay in Supplementary Fig. 2B with quantitation on Fig. 2D, which clearly indicated that Nedd4I cKO and DKO organoids showed increase organoid formation efficiency. This is consistent with the Edu staining data, which together indicate that there are increased proliferative stem cells in the mutant organoids.

Referee #2:

In this manuscript, Novellademunt et al investigated the role of the E3 ligase Nedd4/Nedd4I in normal intestinal crypt homeostasis and colon cancer driven by APCmin. They find that Nedd4/Nedd4I KO in the intestinal epithelium induces the proliferation of intestinal crypts cells and the number of ISCs. They claim that Wnt signaling is elevated in DKO organoids, which undergo reduced apoptosis upon limited addition of Rspodin. They also show that Nedd4/Nedd4I exacerbates intestinal tumor grade and phenotype by APCmin background. Using 293T cell lines to delineate the mechanism, they show that Nedd4/Nedd4I induces ubiquitination of LGR5, leading to proteasomal and lysosomal degradation as well as degradation of Dvl2, thereby

reducing Wnt signaling. The manuscript is well written and the findings are novel (E3 ligase for LGR5) and are well supported. However, there are some critical points need to be addressed to strengthen their study.

Response:

We thank the reviewer's positive comment, and we appreciate the reviewer's concerns and constructive suggestions. These are all relevant to the conclusion of the study, and have now been addressed accordingly in the specific comments below.

Major points

1. A main concern is whether the in vivo effects from DKO can solely be explained by LGR5 regulation. Since the authors also observed Dvl2 degradation by Nedd4/Nedd4I, increased LGR5 or increased Dvl2 or both can cause hyperproliferation of the crypt cells in DKO. To prove that Nedd4/Nedd4I KO increases the proliferation and tumor phenotype via LGR5 regulation, some rescue experiments are needed. For example, if they can rescue the survival rate of APCmin Nedd4/Nedd4I DKO by crossing with LGR5^{+/-}, this would support their hypothesis and also rule out the effect from Dvl2 regulation. However, such a time consuming experiment may be beyond the scope of this study. In its absence, the claim that in vivo effects from DKO are solely via LGR5 regulation should be toned down.

Response:

We appreciate the reviewer's concern about the in vivo phenotype and the complex role of Nedd4/Nedd4I on Dvl2 and Lgr5. We completely agree with the reviewer that it is hard to dissect if the increased ISC and tumour development phenotypes in the DKO animals were caused by Dvl2 or Lgr5 stabilisation or both. We thank the reviewer's suggestion about further breeding with Lgr5 ^{+/-} strain to demonstrate the Lgr5-dependent phenotype, and we appreciate the reviewer's considerate comment that such a time-consuming experiment will be beyond the scope of the current study. In fact, we never meant to claim that the phenotypes were solely caused by Lgr5 stabilisation. Our in vitro data show that Nedd4/Nedd4I can target both DVL2 and LGR5 for degradation, which suggest that the in vivo phenotype could be caused by both. On the other hand, we have generated in vitro data to demonstrate that NEDD4/NEDD4L were able to suppress Wnt signalling in the absence of DVL2 (DVL2 CRISPR KO), indicating that NEDD4/NEDD4L can regulate Wnt signalling in DVL-independent manner. We have now further provided siLGR4/5 data (as suggested by the reviewer in point 4) to demonstrate that the NEDD4/NEDD4L-mediated Wnt regulation is indeed dependent on LGR5 (new Fig. 6F and Supplementary Fig. 6C).

Although our *in vitro* data clearly indicate that *Nedd4/Nedd4l* exhibit *Dvl2*-independent role for *Lgr5* stabilisation and *Wnt* activation, it is still not sufficient to clarify whether the DKO phenotypes were a consequence of *Dvl2* or *Lgr5* stabilisation or both. We apologise for not making this point clear in the original manuscript. We have now improved the discussion to emphasise that the phenotype can be a consequence of both *Dvl2* and *Lgr5* stabilisation.

2. In Figure 1, the author only show *Olfm4* as ISC marker. Multiple ISC markers should be checked such as *LRIG1*, *Ascl2*, *LGR5*.

Response:

We thank the reviewer's suggestion. We have now provided the qRT-PCR analysis of WT and DKO intestinal crypts using the ISC markers suggested by the reviewer (new Supplementary Fig. 1C). Consistent with the *Olfm4* data, other ISC markers such as *Lrig1*, *Ascl2* and *Lgr5* were all upregulated in the DKO crypts, supporting the notion that loss of *Nedd4/Nedd4l* induces ISC expansion.

3. The authors mainly rely on overexpression to prove their mechanism (LGR4/5, *Dvl2*). Can they try to detect endogenous LGR4 or LGR5 and *Dvl2* to show the effect of *Nedd4/Nedd4l* overexpression or knockout?

Response:

We appreciate the reviewer's suggestion for the use of endogenous system to prove the mechanism. We have not done any endogenous experiment previously due to the lack of reliable endogenous LGR4 and LGR5 antibodies. In our hand, we have tried at least 4 different antibodies and none of them offer reliable results on either immunostaining, Western blotting or FACS sorting. We have even obtained customised antibodies generated from Nick Barker's lab from Singapore (who originally discovered Lgr5 as intestinal stem cell marker) and have tested extensively in our lab. Of note, Nick told us that those two antibodies were still in the testing stage and were unlikely to work in endogenous level. Here are FACS results:

Antibody 1: Origene (TA503316) conjugated with Alexa Fluor 647 or APC

Antibodies 2 and 3: Kind gift from Nick Barker's Lab (D2 and G4), unconjugated

Antibody 4: R&D PE-conjugated (FAB8240P)

Overall, none of the antibodies tested can actually detect *Lgr5* protein in either mouse or human samples. The FASC results are the same with or without the primary antibodies, indicating that the signal detected was just non-specific noise. The only antibody that showed increased signal was #4 from R&D (FAB8240P). However, when we further analysed the *Gfp+ve* population from *Lgr5-Gfp* intestine, there was no enrichment at all using this antibody, indicating that the signal was not specific to *Lgr5*.

On the other hand, we have managed to optimise 1 commercially available LGR4 antibody (#sc-390630, Santa Cruz) that gives reliable endogenous LGR4 Western band in the expected size in human samples.

Consistently, our new data show that both LGR4 and DVL2 endogenous protein were stabilised in the NEDD4 and NEDD4L KO cells (new Supplementary Fig. 6B).

4. The evidence that *Nedd4/Nedd4l* regulates Wnt signaling only via LGR5, is weak. To further prove their point *Nedd4/Nedd4l* lof effect in Hek293 cells on Topflash should be compared between Wnt3a only and Wnt3a+Rspo. Furthermore, *Nedd4/Nedd4l* lof Topflash activity should be compared {plus minus} siLGR4,5.

Response:

We thank the reviewer's valuable suggestion to strengthen our findings on the LGR5-dependent role of *Nedd4/Nedd4l* in Wnt regulation. As suggested by the reviewer, we have now compared the *Nedd4/Nedd4l* LOF effect on TOPFlash activity between Wnt3a only and Wnt3a+Rspo. The results showed that Wnt signalling could be activated by *Nedd4/Nedd4l* deletion in either Wnt3a only and Wnt3a+Rspo conditions (new Supplementary Fig. 4B).

The increase in Wnt activity in Wnt3a only condition is likely due to the DVL2-dependent effect. To examine DVL2-independent role, we have previously performed TOPFlash assay in DVL2-KO cells to show that expression of *Nedd4/Nedd4l* was still able to inhibit Wnt signalling in the absence of DVL2 (original Fig. 5D). In addition, we have now further compared the *Nedd4/Nedd4l* LOF effect on TOPFlash activity in plus/minus siLGR4/5 condition as suggested by the reviewer (new Supplementary Fig. 6C and Fig. 6F).

The new results show that loss of LGR4/LGR5 abrogates the Wnt activation mediated by NEDD4/NEDD4L KO, indicating that the NEDD4/NEDD4L-mediated Wnt regulation is indeed dependent on LGR4/5.

As discussed earlier, we agree with the reviewer that the Nedd4/4l regulate Wnt signalling via both Lgr5 and Dvl2. This point is now being clarified in the discussion of the revised manuscript.

5. For the ubiquitination assay in Figure 6, the author show that Nedd4/Nedd4l can ubiquitylate LGR4/5 by pull down of LGR4/5 and detecting HA-UB in cell lysates. To provide more direct evidence for this model, an in vitro ubiquitination assay should be performed with recombinant proteins. At the minimum, pulled-down sample could be boiled with 1% SDS to remove LGR4/5 bound ubiquitinated proteins; dilute 10 times with lysis buffer and pull down it again with the same antibody to detect ubiquitination of LGR4/5. This way, the authors can at least conclude that ubiquitination is not due to other proteins binding to LGR4/5.

Response:

Once again, we thank the reviewer's valuable suggestion on the ubiquitination assay. To further ensure that the ubiquitination observed is of LGR5 itself rather than its associated proteins, we have now repeated the experiment by boiling the pulled-down samples in 1% SDS as suggested by the reviewer. Consistent with our original data, the new results showed that LGR5 ubiquitination was strongly enhanced upon expression of WT NEDD4 or NEDD4L constructs but not the catalytic inactive mutants (new Fig. 6C-6D).

Minor points

1. From RNA scope results, Nedd4 expression is more restricted in the crypts than Nedd4l. However, in many figures, the effect of Nedd4 KO is milder than Nedd4l KO (Figure 2A, C, D. Figure 3C, D. Figure 4A and Supplement 3A and B). Have the author seen the increased Nedd4l level in Nedd4 KO as a compensation? Or is there an explanation for this?

Response:

Indeed, we have noticed the slightly milder phenotype in Nedd4 KO than in Nedd4l KO as indicated by the reviewer. However, we did not see any compensation increase of Nedd4l level in the Nedd4 KO tissues or vice versa (Suppl Fig. 2A). Our current data did show that Nedd4 and Nedd4l are redundant to each other regarding Wnt regulation and mouse phenotypes. However, previous studies have reported several other substrates for Nedd4 and Nedd4l, such as EGFR and Notch receptor (Chen and Matesic, Cancer Metastasis Rev 2007). The additional substrates for Nedd4 and Nedd4l may perhaps explain the differences between Nedd4 KO and Nedd4l KO phenotypes.

2. Comparing Figure 6A and B for LGR5 level, Nedd4 completely removed the LGR5 band from the western blot while LGR5 level seems not be changed upon Nedd4 overexpression from the immunofluorescence picture. If the authors took the picture where there are still significant LGR5 signaling in the cells, it should be mentioned in the text or legends.

Response:

The reviewer questioned about the discrepancy between the Western blot and SNAP data that NEDD4 appeared to completely remove LGR5 band in the Western Blot while LGR5 level was not affected in the SNAP IF images. First, we want to emphasise that the membrane level of LGR5 (mature form) was completely removed by NEDD4 and NEDD4L in the SNAP IF data, while the total LGR5 level (mostly cytoplasmic as immature form in the biosynthetic pathway e.g. ER) was unaffected (Fig. 6B). This suggest that the NEDD4-mediated LGR5 degradation seems to take place at the cell surface and makes the mature (SNAP-labelled) form disappear. It is indeed interesting to note that LGR5 appears to be completely degraded by NEDD4/4L in the Western Blot (Fig. 6A). However, if we take a closer look at the blot, there are indeed two bands for LGR5-Flag protein. The lower band (indicated by white triangle) is likely the immature ER form and the higher band (indicated by black triangle) is the mature, glycosylated post-Golgi form (new Fig. 6A).

The higher band (mature form) is a lot more prominent than the lower band (immature form) in our LGR5-Flag construct. And the degradation of LGR5 happens largely to the higher mature form than the lower immature form, which is consistent to the SNAP IF data (predominant degradation in the mature form at the membrane level of mature LGR5). We thank the reviewer for pointing this out. We have now modified Fig. 6A and have clarified this point in the revised results session.

3. Figure legend for Figure 6B is not matched with Figure 6B. Please check the Figure legend in the corresponding supplement figure.

Response:

We apologise for the mistake. This has now been corrected in the revised manuscript.

4. Nedd4/Nedd4l overexpression in 293T cell or using Nedd4 or Nedd4l KO 293T cell lines, the authors need to prove mRNA levels of LGR4, 5 are not changed.

Response:

We have now examined the mRNA level of LGR4 and LGR5 in the NEDD4 and NEDD4L KO 293T cells as suggested by the reviewer. Indeed, we did not observe any transcriptional changes of the LGR receptors in NEDD4/4L KO, indicating that the protein changes were caused by post-translational modification.

5. In Figure 5 E-G, the authors showed that Nedd4/Nedd4l overexpression decreased Topflash in HCT116 but not in APC4 and DLD1. However, there is a study (PMID : 23349017; Al-Kharusi et al, 2013, Carcinogenesis) showing that there is no detectable LGR5 protein in HCT116. Thus, it is worth to show that LGR5 is expressed in HCT116, APC4, DLD-1 (qPCR or western blot).

Response:

The reviewer asked if LGR5 is expressed in the various cancer cell lines that we have examined. We have now performed qPCR analysis to show that LGR5 was expressed at similar level in HEK293T, APC4 and HCT116 cells, while expression in DLD1 was significantly increased. The CT values of LGR5 in HCT116 (29.5) was comparable to HEK293T (28.8) cells, indicating that LGR5 is indeed expressed in HCT116 cells.

We want to emphasise that all the Lgr5 antibodies that we have tested, including the one in the Al-Kharusi et al 2013 as pointed out by the reviewer, can never detect endogenous LGR5 protein reliably. The lack of protein being detected by Western blotting may not be the most reliable way to confirm the expression in the case of Lgr5.

2nd Editorial Decision

8th Nov 2019

Thank you for submitting your revised manuscript for consideration by The EMBO Journal. Your amended study was sent back to the referees for re-evaluation, and we have received comments from both of them, which I enclose below.

As you will see the referee finds that their concerns have been sufficiently addressed and they are now broadly in favour of publication.

Thus, we are pleased to inform you that your manuscript has been accepted in principle for publication in The EMBO Journal, pending some minor issues related to formatting and data representation as listed below, which need to be adjusted at re-submission.

REFEREE REPORTS:

Referee #1:

The authors have now addressed most of my previous concerns regarding the original presented results and the new required experiments (or at least explained why they cannot be done, especially the proof that endogenous LGR5 is stabilized in vivo in mice that lack Nedd4/Nedd4L). This is commendable, as they performed and show numerous new experiments. The novelty of the work described in this paper in general is still somewhat in question, however.

Referee #2:

The revised manuscript from Novellasdemunt et al is substantially improved. This manuscript aims to study the role of Nedd4/Nedd4L E3 ligases in intestinal stem cell homeostasis and cancer prone situation. They focused on regulation of Dvl2, a previously identified substrate of Nedd4/Nedd4L, as well as a newly discovered substrate, LGR5. They show that loss of Nedd4/Nedd4L increases proliferation of the intestinal epithelium and Wnt signaling. They show convincingly that LRG4/5 levels are regulated by Nedd4/Nedd4L. The authors also showed that Nedd4/Nedd4L induces ubiquitination and proteasomal degradation of LGR5. The main concerns raised in the review and how these have been addressed are summarized below.

1) In vivo effect of Nedd4/Nedd4L DKO solely through LGR5 regulation: They toned down the statement that the effect of DKO could be explained either by LGR5 or Dvl2 or both.

- 2) Multiple ISC markers should be checked: They addressed this by qPCR showing that Lgr1, Ascl1, and Lgr5 expressions were increased in DKO.
- 3) Effect of Nedd4/Nedd4l on endogenous LGR4/5 or Dvl2: They tried many antibodies to address this and show the compelling evidence that there is no working antibodies available for LGR5. For LGR4 and Dvl2, they showed that Nedd4/Nedd4l KO increase endogenous LGR4 and Dvl2 protein levels in cells. Moreover, they generated endogenously tagged LGR5-HA Knock-In organoids from WT and DKO and showed that LGR5 and Dvl2 protein levels are increased in DKO organoids. They also provide endogenous interaction between Nedd4/Nedd4l and LGR4 or Dvl2.
- 4) The evidence that Nedd4/Nedd4l regulates Wnt signaling only via LGR5 is weak: The authors tried to address this by Topflash. They showed that Nedd4/Nedd4l regulate Wnt signaling via LGR5 and Dvl2 and this has been clarified in the discussion.
- 5) Ubiquitination assay for LGR5: The author addressed this with a modified protocol.
- 6) Discrepancy between Western blot and SNAP upon Nedd4/Nedd4l overexpression: The author argue that there are two band for LGR5 (Upper band for mature and lower band for immature forms of LGR5) and Nedd4/Nedd4l overexpression predominantly remove mature form of LGR5 from Western blot and SNAP. However, in Figure 6A, there is same lower band in EV transfected control lane. Also, the similar band could be seen in Ctr lane in Figure 6E. Therefore, what the authors claimed as immature LGR5 band could be a nonspecific band unless they perform additional experiments (Surface biotinylation assay or EndoH assay).
- 7) Quantification, Statistics: They included densitometry quantification for cyclohexamide experiment and showed the quantification for SNAP data.

Overall, the authors have addressed most major and minor points from referees with compelling data or reasonable explanations. With this revised manuscript, I feel now this can be published in EMBO J.

2nd Revision - authors' response

25th Nov 2019

The authors performed the requested editorial changes.

3rd Editorial Decision

28th Nov 2019

Thank you for submitting the revised version of your manuscript. I have now evaluated your amended manuscript and concluded that the remaining minor concerns have been sufficiently addressed.

Thus, I am pleased to inform you that your manuscript has been accepted for publication in the EMBO Journal.

Corresponding Author Name: Vivian Li

Journal Submitted to: THE EMBO Journal

Manuscript Number: EMBOJ-2019-102771